# Mapping mutational effects along the evolutionary landscape of HIV envelope

**Hugh K Haddox[1,2†], Adam S Dingens[1,2†], Sarah K Hilton[1,3], Julie Overbaugh[4,5], Jesse D Bloom[1,3]***

[1]Basic Sciences Division and Computational Biology Program, Fred Hutchinson Cancer Research Center, Seattle, United States; [2]Molecular and Cellular Biology PhD program, University of Washington, Seattle, United States; [3]Department of Genome Sciences, University of Washington, Seattle, United States; [4]Human Biology Division, Fred Hutchinson Cancer Research Center, Seattle, United States; [5]Epidemiology Program, Fred Hutchinson Cancer Research Center, Seattle, United States

**Abstract** The immediate evolutionary space accessible to HIV is largely determined by how single amino acid mutations affect fitness. These mutational effects can shift as the virus evolves. However, the prevalence of such shifts in mutational effects remains unclear. Here, we quantify the effects on viral growth of all amino acid mutations to two HIV envelope (Env) proteins that differ at >100 residues. Most mutations similarly affect both Envs, but the amino acid preferences of a minority of sites have clearly shifted. These shifted sites usually prefer a specific amino acid in one Env, but tolerate many amino acids in the other. Surprisingly, shifts are only slightly enriched at sites that have substituted between the Envs—and many occur at residues that do not even contact substitutions. Therefore, long-range epistasis can unpredictably shift Env's mutational tolerance during HIV evolution, although the amino acid preferences of most sites are conserved between moderately diverged viral strains.
DOI: https://doi.org/10.7554/eLife.34420.001

***For correspondence:**
jbloom@fredhutch.org

[†]These authors contributed equally to this work

**Competing interests:** The authors declare that no competing interests exist.

## Introduction

HIV's envelope (Env) protein evolves very rapidly. The major group of HIV-1 that is responsible for the current pandemic originated from a virus that entered the human population ~100 years ago (*Sharp and Hahn, 2011*; *Worobey et al., 2008*; *Faria et al., 2014*). The descendants of this virus have evolved so rapidly that their Envs now have as little as 65% protein identity (*Lynch et al., 2009*). For comparison, protein orthologs shared between humans and mice have only diverged to a median identity of 78% over 90 million years (*Waterston et al., 2002*; *Hedges et al., 2006*).

Env's rapid evolution has dire consequences for anti-HIV immunity, since it erodes the efficacy of most neutralizing antibodies (*Albert et al., 1990*; *Wei et al., 2003*; *Richman et al., 2003*; *Burton et al., 2005*). Because of this public-health importance, numerous studies have experimentally characterized aspects of the 'evolutionary landscape' that Env traverses. The immediate evolutionary space accessible to any given Env is largely defined by the effects on viral fitness of all single amino acid mutations to Env. Most mutational studies have measured how just a small number of these mutations affect viral growth in cell culture, although it has recently become possible to use deep mutational scanning to measure the effects of many (*Al-Mawsawi et al., 2014*; *Duenas-Decamp et al., 2016*) or even all (*Haddox et al., 2016*) single amino acid mutations to an Env variant.

But interpreting these studies in the context of Env evolution requires addressing a fundamental question: How informative are mutational studies of a single protein variant about constraints on

**eLife digest** The virus that causes AIDS, or HIV, has a protein called Env on its surface, which is essential for the virus to infect cells. Env can also be recognized by the immune system, which then targets the virus for destruction or blocks it from infecting cells. Unfortunately, Env evolves very quickly, which means that HIV can evade our defenses. However, there are limits to how much this protein can change, since it still needs to perform its essential role in helping viruses enter cells.

In the century since HIV first appeared in human populations, the virus has evolved considerably. There are now many HIV strains that infect people, and they bear Env proteins with substantially different sequences. However, it is not clear if these changes in sequence have resulted in Envs from distinct strains being able to tolerate different mutations.

To examine this question, Haddox et al. compared how the Envs from two strains of HIV react to modifications in their sequences. They created all possible individual mutations in the proteins, and the resulting collections of mutated viruses were then tested for their ability to infect cells in the laboratory.

Most mutations had similar effects in both Env proteins. This allowed Haddox et al. to identify portions of the protein that easily accommodate changes, and portions that must remain unchanged for viruses to remain infectious—at least in the laboratory. Some of these mutations are under different types of pressures when the virus faces the immune system, and those were identified using computational approaches.

However, some mutations were tolerated differently by the two Env proteins. Therefore, viral strains differ in how their Env proteins can evolve. The parts of Env that showed differences in mutational tolerance between the strains were not necessarily the parts that differ in sequence. This shows that changes in sequence in one part of the protein can modify how other portions evolve.

It remains to be determined whether changes in tolerance to mutations translate into differences in how the virus can escape immunity. This is an important question given that the rapid evolution of Env is a major obstacle to creating a vaccine for HIV.

DOI: https://doi.org/10.7554/eLife.34420.002

long-term evolution? During protein evolution, substitutions at one site can change the effect of mutations at other sites (*Natarajan et al., 2013*; *Gong et al., 2013*; *Harms and Thornton, 2014*; *Podgornaia and Laub, 2015*; *Starr and Thornton, 2016*; *Klink and Bazykin, 2017*). We will follow the nomenclature of (*Pollock et al., 2012*) to refer to these changes in mutational effects as *shifts* in a site's amino acid preferences. Such shifts can accumulate as substitutions become entrenched via epistatic interactions with subsequent changes (*Starr et al., 2017*; *Pollock et al., 2012*; *Shah et al., 2015*; *Bazykin, 2015*)—although the magnitude of these shifts is usually limited (*Doud et al., 2015*; *Chan et al., 2017*; *Ashenberg et al., 2013*; *Risso et al., 2015*).

Given that the Envs of circulating HIV strains represent a vast collection of homologs that often differ at >100 residues, shifts in amino acid preferences could make the outcome of any study highly dependent on the Env used. Indeed, epistasis among a few combinations of Env mutations has been experimentally demonstrated (*da Silva et al., 2010*), and epistatic fitness landscapes have been computationally inferred for a variety of HIV proteins (*Kouyos et al., 2012*; *Ferguson et al., 2013*; *Mann et al., 2014*; *Barton et al., 2015*) including Env (*Louie et al., 2018*). However, the only protein-wide experimental studies of how amino acid preferences shift during evolution have examined proteins that are structurally far simpler than Env, which forms a large heavily glycosylated heterotrimeric complex that transitions through multiple conformational states (*Munro et al., 2014*; *Ozorowski et al., 2017*).

Here, we use an improved version of a previously described deep mutational scanning strategy (*Haddox et al., 2016*) to measure the effects on viral growth of all single amino acid mutations to two transmitted-founder virus Envs that differ by >100 mutations. We compare these complete maps of mutational effects to identify sites that have shifted in their amino acid preferences between the Envs. Most sites show no detectable shifts, but 30 sites have clearly shifted preferences. These shifted sites usually prefer a specific amino acid in one Env but have shifted to tolerate many amino acids in the other Env. The shifted sites cluster in structure but are often distant from any amino acid

substitutions that distinguish the two Envs, demonstrating the action of long-range epistasis. By aggregating our measurements for both Envs, we identify sites that evolve faster or slower in nature than expected given the functional constraints measured in the lab, probably due to pressure for immune evasion. Overall, our work provides complete across-strain maps of mutational effects that inform analyses of Env's evolution and function.

## Results

### Two Envs from clade A transmitted-founder viruses

The viruses most relevant to HIV's long-term evolution are those which are transmitted from human-to-human. However, the only prior work that has measured how all Env amino acid mutations affect HIV growth is a study by some of us (*Haddox et al., 2016*) that used a late-stage lab-passaged CXCR4-tropic virus (*Peden et al., 1991*). The properties of Env can vary substantially between such late-stage viruses and the transmitted-founder viruses relevant to HIV's long-term evolution (*Sagar et al., 2006*; *Wilen et al., 2011*; *Parrish et al., 2013*; *Ronen et al., 2015*).

We therefore selected Envs from two transmitted-founder viruses, BG505.W6M.C2.T332N and BF520.W14M.C2 (hereafter referred to as BG505 and BF520), that were isolated from HIV-infected infants shortly after mother-to-child transmission (*Nduati et al., 2000*; *Wu et al., 2006*; *Goo et al., 2014*). The BG505 Env has been extensively studied from a structural standpoint (*Julien et al., 2013*; *Lyumkis et al., 2013*; *Pancera et al., 2014*; *Huang et al., 2014*; *Sanders et al., 2015*; *Stewart-Jones et al., 2016*; *Gristick et al., 2016*), and variants of this Env are being tested as vaccine immunogens (*Sanders et al., 2013*, *Sanders et al., 2015*; *de Taeye et al., 2015*). We used the T332N variant of BG505 Env because it has a common glycosylation site that is targeted by many anti-HIV antibodies (*Sanders et al., 2013*). The BF520 Env was isolated from an infant who developed an early broad anti-HIV antibody response (*Goo et al., 2014*; *Simonich et al., 2016*). We have previously created comprehensive codon-mutant libraries of the BF520 Env and used them to map HIV antibody escape (*Dingens et al., 2017*), but these BF520 libraries have not been characterized with respect to how mutations affect viral growth.

Both BG505 and BF520 are from clade A of the major (M) group of HIV-1. *Figure 1* shows the phylogenetic relationship among these two Envs and other clade A sequences. BG505 and BF520 are identical at 721 of the 836 pairwise-alignable protein sites (86.2% identity). However, in our experiments we mutagenized only the ectodomain and transmembrane domain of Env, and excluded the signal peptide and cytoplasmic tail. The reason is that we measure how Env mutations affect viral growth, which is influenced both by the functionality of Env protein molecules and their expression level. Mutations in the signal peptide and cytoplasmic tail commonly affect Env expression level (*Chakrabarti et al., 1989*; *Yuste et al., 2004*; *Li et al., 1994*), so we excluded these regions with the goal of reducing the degree to which we simply identified mutations that affected Env expression. In the ectodomain and transmembrane domains of Env, BG505 and BF520 are identical at 549 of the 616 sites (89.1% identity) that are alignable across clade A Envs (*Figure 1—source data 1*, *Figure 1—source data 2*). The divergence between BG505 and BF520 therefore offers ample opportunity to investigate mutational shifts during Env evolution.

### Deep mutational scanning of each Env

We have previously described a deep mutational scanning strategy for measuring how all amino acid mutations to Env affect HIV growth in cell culture, and applied this strategy to the late-stage lab-adapted LAI strain (*Haddox et al., 2016*). Here, we made several modifications to this earlier strategy to apply it to transmitted-founder Envs and to reduce the experimental noise. This last consideration is especially important when comparing Envs, since it is only possible to reliably detect differences that exceed the magnitude of the experimental noise. Our modified deep mutational scanning strategy is in *Figure 2A*. This approach had the following substantive changes: instead of SupT1 cells, we used SupT1.CCR5 cells (SupT1 cells that express CCR5 in addition to CXCR4 [*Boyd et al., 2015*]) to support growth of viruses with transmitted-founder, CCR5-tropic Envs; we used more virions for the first passage ($\geq 3 \times 10^6$ versus $5 \times 10^5$ infectious units per library) to avoid bottlenecking library diversity; and rather than performing a full second passage we just did a short high-MOI infection to enable recovery of *env* genes from infectious virions without bottlenecking

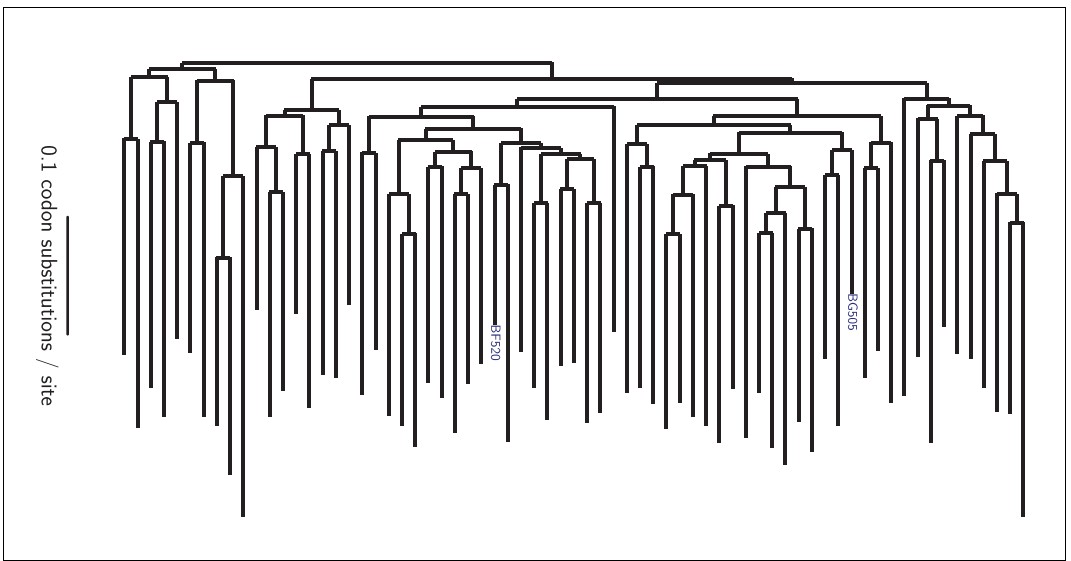

**Figure 1.** Phylogenetic tree showing the relationship of BG505 and BF520 to other clade A Envs. The tree shows the 69 Envs in the alignment in *Figure 1—source data 1*, which is a subsample of clade A sequences from the group M alignment in the Los Alamos HIV sequence database (http://www.hiv.lanl.gov). Sites not mutagenized in our experiments (the signal peptide and cytoplasmic tail) or that are poorly alignable were masked as indicated in *Figure 1—source data 2*, leaving 616 alignable sites. The pairwise identity of BG505 and BF520 to other sequences at alignable sites is in *Figure 1—figure supplement 1*. The tree topology was inferred using RAxML (*Stamatakis, 2014*) under the GTRCAT model of nucleotide substitution, and branch lengths were optimized under the M0 Goldman-Yang model (*Yang et al., 2000*) using phydms (*Hilton et al., 2017*).

DOI: https://doi.org/10.7554/eLife.34420.003

The following source data and figure supplement are available for figure 1:

**Source data 1.** The alignment of clade A *env* coding sequences is in cladeA_alignment.fasta.

DOI: https://doi.org/10.7554/eLife.34420.004

**Source data 2.** The 240 Env sites masked in all phylogenetic analyses because they were not mutagenized in our experiments or are poorly alignable are listed in alignment_mask.csv.

DOI: https://doi.org/10.7554/eLife.34420.005

**Figure supplement 1.** Pairwise identity of all Env sequences to BG505 and BF520.

DOI: https://doi.org/10.7554/eLife.34420.006

---

(*Figure 2A*). We performed this deep mutational scanning in full biological triplicate for both BG505 and BF520 (*Figure 2B*). Our libraries encompassed all codon mutations to all sites in Env except for the signal peptide and cytoplasmic tail.

The deep mutational scanning effectively selected for functional Envs as evidenced by strong purifying selection against stop codons. *Figure 3A* shows the average frequency of mutations across Env in the plasmid mutant libraries, the mutant viruses, and wildtype controls as determined from the deep sequencing. The mutant viruses show clear selection against stop codons and many nonsynonymous mutations (*Figure 3A*). This selection is more apparent if we correct for the background error rates estimated from the wild-type controls (*Figure 3—source data 1*). The error-corrected frequencies of stop codons drop to 3–16% of their original values (*Figure 3—source data 1*), with the residual stop codons probably due to some non-functional virions surviving due to complementation by other co-infecting virions. The error-corrected frequencies of nonsynonymous mutations also drop substantially (43%–49% of their original values), whereas the frequencies of synonymous mutations drop only slightly (85%–95% of their original values). These trends are consistent with the fact that nonsynonymous mutations are often deleterious, whereas synonymous mutations often (*Zanini and Neher, 2013*) have only mild effects on viral growth. *Figure 3A* only summarizes one aspect of the deep mutational scanning data, but *Supplementary file 1* and *2* contain detailed plots showing all aspects of the data (read depth, per-site mutation rate, etc) as generated by the dms_tools2 software (*Bloom, 2015*, https://jbloomlab.github.io/dms_tools2/).

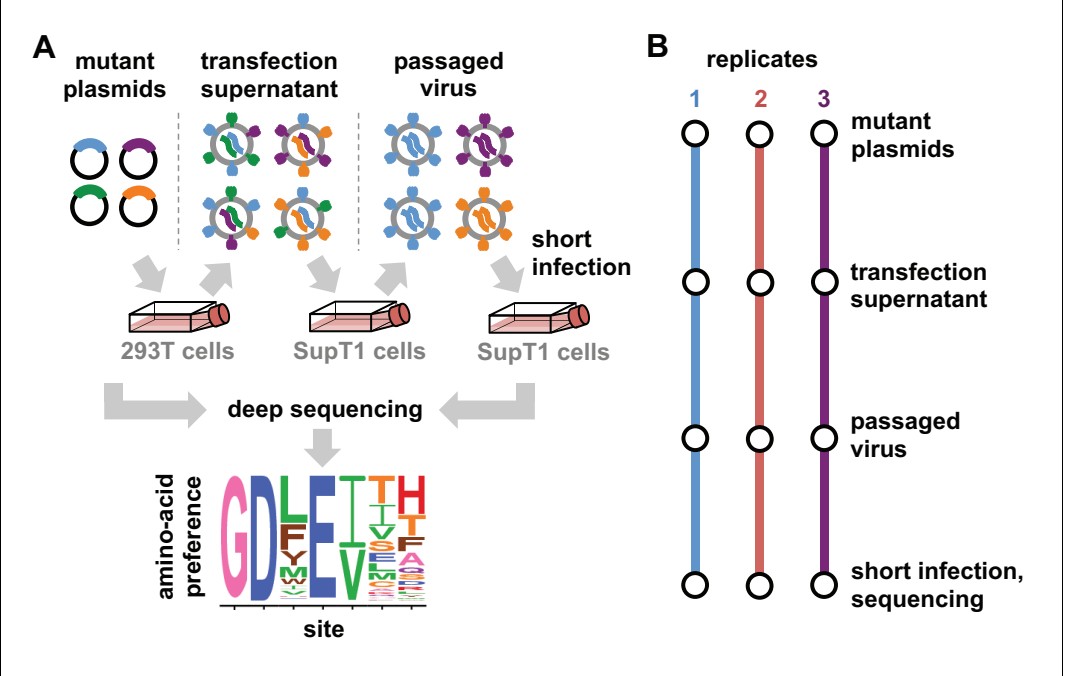

**Figure 2.** Deep mutational scanning workflow. (**A**) We made libraries of proviral HIV plasmids with random codon-level mutations in the *env* gene. The number of mutations per gene approximately followed a Poisson distribution with a mean between 1 and 1.5 (***Figure 2—figure supplement 1***). We transfected the plasmids into 293T cells to generate mutant viruses, which lack a genotype-phenotype link since cells are multiply transfected. To establish a genotype-phenotype link and select for Env variants that support HIV growth, we passaged the libraries in SupT1. CCR5 cells for four days at a low multiplicity of infection (MOI) of 0.01. To isolate the env genes from only viruses that encoded a functional Env protein, we infected the passaged libraries into SupT1.CCR5 cells at high MOI and harvested reverse-transcribed non-integrated viral DNA after 12 hr. We then deep sequenced the env genes from these final samples as well as the initial plasmid library, using molecular barcoding to reduce sequencing errors. We also deep sequenced identically handled wildtype controls to estimate error rates. Using these sequencing data, we estimated the preference for each of the 20 amino acids at each site in Env. These data are represented in logo plots, with the height of each letter proportional to that site's preference for that amino acid. (**B**) We conducted this experiment in full biological triplicate for both BG505 and BF520, beginning each replicate with independent creation of the plasmid mutant library. These replicates therefore account for all sources of noise and error in the experiments.

DOI: https://doi.org/10.7554/eLife.34420.007

The following figure supplement is available for figure 2:

**Figure supplement 1.** Sanger sequencing of selected clones from the mutant plasmid libraries.
DOI: https://doi.org/10.7554/eLife.34420.008

We used the deep mutational scanning data to estimate the preference of each site in Env for each amino acid via the analysis method described in ***Bloom, 2015***). As graphically illustrated in ***Figure 2A***, the preferences for each site are normalized to sum to one. Our libraries were mutagenized at 670 sites in BG505 and 662 sites in BF520, so $670 \times 20 = 13,400$ and $662 \times 20 = 13,240$ preferences were estimated for each Env, respectively. The correlations between the preferences from different experimental replicates are in ***Figure 3B***, and the preferences themselves are in ***Figure 3—source data 2***. These replicate-to-replicate correlations are substantially higher than those for the deep mutational scanning of LAI Env by (***Haddox et al., 2016***), which had replicate-to-replicate Pearson correlations of only $R = 0.45$ to $0.50$.

While the replicates are well correlated across all replicates for both BG505 and BF520, the replicates for BG505 are more correlated with each other than with replicates for BF520, and vice versa (***Figure 3B***, compare red and blue versus gray plots). This fact hints that there are some shifts in amino-acid preferences between the two Envs—something that is investigated with more statistical rigor later in this paper. Note also that there is a trend for highly preferred amino acids to be more

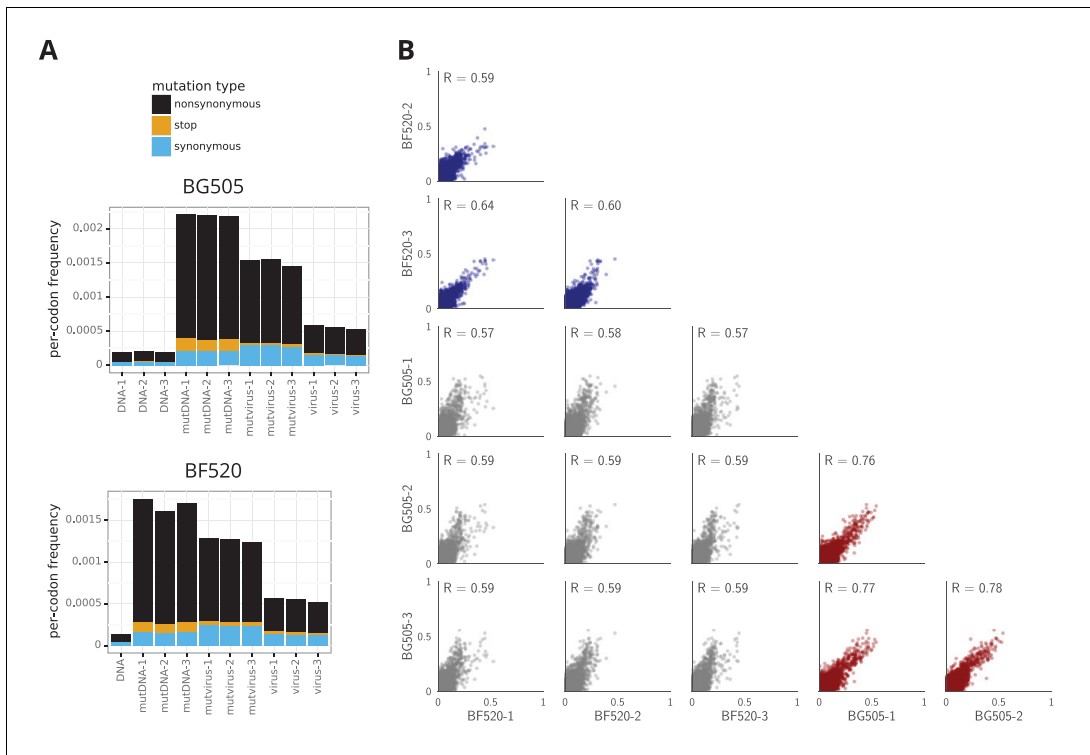

**Figure 3.** The deep mutational scanning selects for functional Envs and yields measurements that are well correlated among replicates. (**A**) The average per-codon mutation frequency when sequencing plasmids encoding wildtype Env (*DNA*), plasmid mutant libraries (*mutDNA*), mutant viruses after the final infection (*mutvirus*), and virus generated from wild-type plasmids (*virus*). Mutations are categorized as nonsynonymous, synonymous, or stop codon. The *DNA* samples show that sequencing errors are rare, and the *virus* samples show that viral-replication errors are well below the frequency of mutations in the *mutDNA* samples. Comparing the *mutvirus* to *mutDNA* shows clear purifying selection against stop codons and some nonsynonymous mutations, particularly after subtracting the background error rates given by the *virus* and *DNA* samples (*Figure 3—source data 2*). More extensive plots from the analysis of the deep sequencing data are in *Supplementary file 1* and *2*. (**B**) Correlations between replicates in the measured preferences of each site in Env for all 20 amino acids. Blue indicates replicate measurements on BF520, red indicates replicate measurements on BG505, and gray indicates across-Env measurements of BF520 versus BG505. *R* is the Pearson correlation coefficient. The numerical values for the preferences are in *Figure 3—source data 2*. *Figure 3—figure supplement 1* shows the correlations using contour rather than scatter plots.

DOI: https://doi.org/10.7554/eLife.34420.009

The following source data and figure supplement are available for figure 3:

**Source data 1.** Average frequencies of nonsynonymous, synonymous, and stop-codon mutations as plotted in mutfreqs are in avgmutfreqs.csv.

DOI: https://doi.org/10.7554/eLife.34420.010

**Source data 2.** Preferences for each replicate and averages are in all_prefs_unscaled.zip.

DOI: https://doi.org/10.7554/eLife.34420.011

**Figure supplement 1.** Correlations plotted on a contour rather than a scatter plot.

DOI: https://doi.org/10.7554/eLife.34420.012

strongly preferred in BG505 than BF520 (most high-preference points in the gray plots in *Figure 3B* fall above the diagonal); however, this trend does not necessarily reflect differences between the Envs. Rather, there were modest differences in the stringency of selection between our deep mutational scans of BG505 and BF520 (*Figure 3—source data 1* shows that purifying selection better purged stop codons in BG505). In the next section, we correct for these experimental differences by calibrating each dataset to match the stringency of selection in nature.

## Amino acid preferences of the Envs and their relationship to HIV evolution

The most immediate question is how authentically the experimental measurements describe the actual selection on Env function in nature. Direct comparisons between experimentally measured amino acid preferences and amino acid frequencies in natural sequences are confounded by the fact that the natural sequences are evolutionarily related. This problem can be overcome by making the comparison in a phylogenetic context to account for the evolutionary relationships among sequences.

Specifically, we used our deep mutational scanning data to construct experimentally informed codon models (ExpCM's) for Env's evolution. An ExpCM is a phylogenetic substitution model that incorporates the functional constraints measured in a deep mutational scanning experiment (*Hilton et al., 2017*). If the experiment captures much of the actual evolutionary constraint on a gene, then an ExpCM will describe the gene's natural evolution better than a standard phylogenetic codon substitution model. The reason is that standard codon substitution models (*Yang et al., 2000*) only model functional constraint via a single parameter that represents the rate of fixation of nonsynonymous protein-altering mutations relative to synonymous ones; this parameter is called dN/dS or $\omega$. In contrast, an ExpCM accounts for the preference of each site for each of the 20 amino acids under the functional selection in the deep mutational scan, and then additionally adds an $\omega$ parameter that represents the relative rate of nonsynonymous to synonymous substitutions after accounting for these functional constraints (*Bloom, 2017*; *Hilton et al., 2017*). Importantly, since we expect some sites in Env to be under diversifying selection from immunity, we extended the ExpCM's described in *Hilton et al., 2017*) to draw $\omega$ from a gamma distribution as is commonly done for codon-substitution models (*Yang et al., 2000*).

*Table 1* shows that ExpCM's informed by the deep mutational scanning of either BG505 or BF520 describe the natural evolution of Env vastly better than a standard codon substitution model. In addition to the improved fit of the ExpCM's, we can also interpret the $\omega$ parameter. Recall that for standard codon substitution models, $\omega$ is simply the rate of fixation of nonsynonymous mutations relative to synonymous ones. For such models, the gene-wide average $\omega$ is almost always <1, since purifying selection purges many functionally deleterious amino acid mutations even for adaptively evolving proteins (*Murrell et al., 2015*). Indeed, *Table 1* shows that Env's gene-wide average $\omega$ is < one for a standard model. But for ExpCM's, $\omega$ is the relative rate of nonsynonymous to synonymous substitutions *after* accounting for functional constraints measured in the deep mutational scanning (*Bloom, 2017*). For the ExpCM's, the gene-wide average $\omega$ is >1 (*Table 1*), indicating that external selection (e.g. from immunity) drives Env to fix amino acid mutations faster than expected under a null model that only accounts for functional constraints on the protein.

ExpCM's also have a stringency parameter that relates selection in the experiments to that in nature. Essentially, this parameter indicates how strongly natural selection prefers the amino acids that are preferred in the deep mutational scanning (*Hilton et al., 2017*). A stringency parameter >1 indicates that natural selection prefers the same amino acids as the experiments, but with greater stringency. Both ExpCM's have stringency parameters >1 (*Table 1*)—a finding that makes sense, since the stop-codon analysis in the previous section suggests that the experimental selections are more lax than natural selection on HIV.

For the entire rest of the paper, we use the experimentally measured preferences re-scaled by the stringency parameters in *Table 1*. The reason we do this is to distinguish genuine differences between the two Envs from mere variation in the strength of selection between the two sets of experiments. Re-scaling both sets of preferences to optimally describe Env evolution in nature is a principled way to standardize the measurements; see (*Hilton et al., 2017*) and the Materials and methods section entitled 'Re-scaling the preferences' for a more detailed explanation.

A qualitative way to assess if the deep mutational scanning authentically describes selection on Env function is to visually compare the measurements with existing knowledge. *Figure 4* and *Figure 5* show the re-scaled across-replicate average of the amino acid preferences for each Env. At sites of known functional importance, these preferences are usually consistent with prior knowledge. For instance, residues T257, D368, E370, W427, and D457 are important for Env binding to CD4 (*Olshevsky et al., 1990*), and all these amino acids are highly preferred in our deep mutational scanning (*Figure 4* and *Figure 5*). Likewise, Env has 10 disulfide bonds (linking sites 54–74, 119–205,

**Table 1.** Evolutionary models informed by the deep mutational scanning describe HIV's evolution in nature much better than a standard substitution model.

Shown are the results of maximum likelihood fitting of substitution models to the clade A phylogeny in tree. Experimentally informed codon models (*Hilton et al., 2017*) utilizing the across-replicate average of the deep mutational scanning describe Env's natural evolution far better than a standard codon substitution model (*Yang et al., 2000*) as judged by comparing the Akaike information criteria (*Posada and Buckley, 2004*). Both ExpCM's have a stringency parameter >1. All models draw $\omega$ from a gamma distribution, and the table shows the mean ($\bar{\omega}$) and shape parameters ($\omega_\alpha$ and $\omega_\beta$) of this distribution. The last two columns show the number of sites evolving faster ($\omega_r > 1$) or slower ($\omega_r < 1$) than expected at a false discovery rate of 0.05, as determined using the approach in *Bloom, 2017* (see also the last section of the Results). Analyses were performed using phydms (*Hilton et al., 2017*, http://jbloomlab.github.io/phydms/). *Table 1—source data 1.* shows the results for additional substitution models.

| Model | $\Delta$AIC | LogLikelihood | nParams | Stringency | $\bar{\omega}$ | $\omega_\alpha$ | $\omega_\beta$ | Nsites $\omega_r > 1$ | Nsites $\omega_r < 1$ |
|---|---|---|---|---|---|---|---|---|---|
| ExpCM BF520 | 0.0 | −35218.8 | 7 | 2.8 | 1.4 | 1.0 | 0.7 | 66 | 35 |
| ExpCM BG505 | 269.0 | −35353.3 | 7 | 2.1 | 1.3 | 0.9 | 0.7 | 65 | 53 |
| Goldman-Yang M5 | 3455.1 | −36941.4 | 12 | nan | 0.8 | 0.6 | 0.7 | 14 | 211 |

DOI: https://doi.org/10.7554/eLife.34420.013

The following source data is available for Table 1:

Source data 1. Results for phylogenetic models where $\omega$ is not drawn from a gamma-distribution or where the preferences are averaged across sites to eliminate the site specificity are in modelcomparison.md.
DOI: https://doi.org/10.7554/eLife.34420.014

126–196, 131–157, 218–247, 228–239, 296–331, 378–445, 385–418, and 598–604), most of which are important for function (*van Anken et al., 2008*)—and the cysteines at these sites are highly preferred in our deep mutational scanning. The deep mutational scanning is also consistent with prior knowledge about sites that are tolerant of mutations. For instance, Env has five variable loops that mostly evolve under weak constraint in nature (*Starcich et al., 1986*; *Zolla-Pazner and Cardozo, 2010*)—and most sites in these loops are mutationally tolerant in our deep mutational scanning (see sites indicated by gray overlay bars in *Figure 4* and *Figure 5*, such as 132 to 195). It is beyond the scope of this paper to catalog associations between our measurements and all other prior mutational studies of Env, but the concordance of our findings with the above mutational studies, and the fact that our data improve phylogenetic models of Env's natural evolution, suggest that our experiments do a reasonable job of authentically measuring functional selection on Env.

## Shifts in amino acid preferences between BG505 and BF520

The most fundamental question that we seek to address is how similar the amino acid preferences are between the two Envs. We have already noted that *Figure 3B* shows that the preferences are more correlated for replicate measurements on the same Env than for replicate measurements on different Envs. However, simply comparing correlation coefficients does not identify specific sites where mutational effects have shifted, nor does it quantify the magnitude of any shifts.

We therefore used a more rigorous approach to identify sites where the amino acid preferences differ between BG505 and BF520 by an amount that exceeds the noise in our experiments. We first re-scaled the preferences from each experimental replicate by the stringency parameter for that Env from *Table 1* to calibrate all measurements to the stringency of natural selection. We then identified the 659 sites in the mutagenized regions of Env that are pairwise alignable between BG505 and BF520 (*Figure 6—source data 1*). For each site, we calculated the shift in amino acid preferences between Envs using an approach similar to that of (*Doud et al., 2015*) as illustrated in *Figure 6A*. This approach calculates the magnitude of the shift after correcting for experimental noise by comparing the differences in preferences between replicates for BG505 and BF520 to the differences between replicates for the same Env. *Figure 6A* shows this calculation for a site that has not shifted (site 598, which strongly prefers cysteine in both Envs), the most shifted site (512, which shifts from being mutationally tolerant in BG505 to strongly preferring alanine in BF520), and two other sites with more intermediate behaviors.

The overall distribution of shifts between BG505 and BF520 is shown in *Figure 6B*. Most sites have relatively small shifts (close to zero), although there is a long tail of sites with large shifts. This

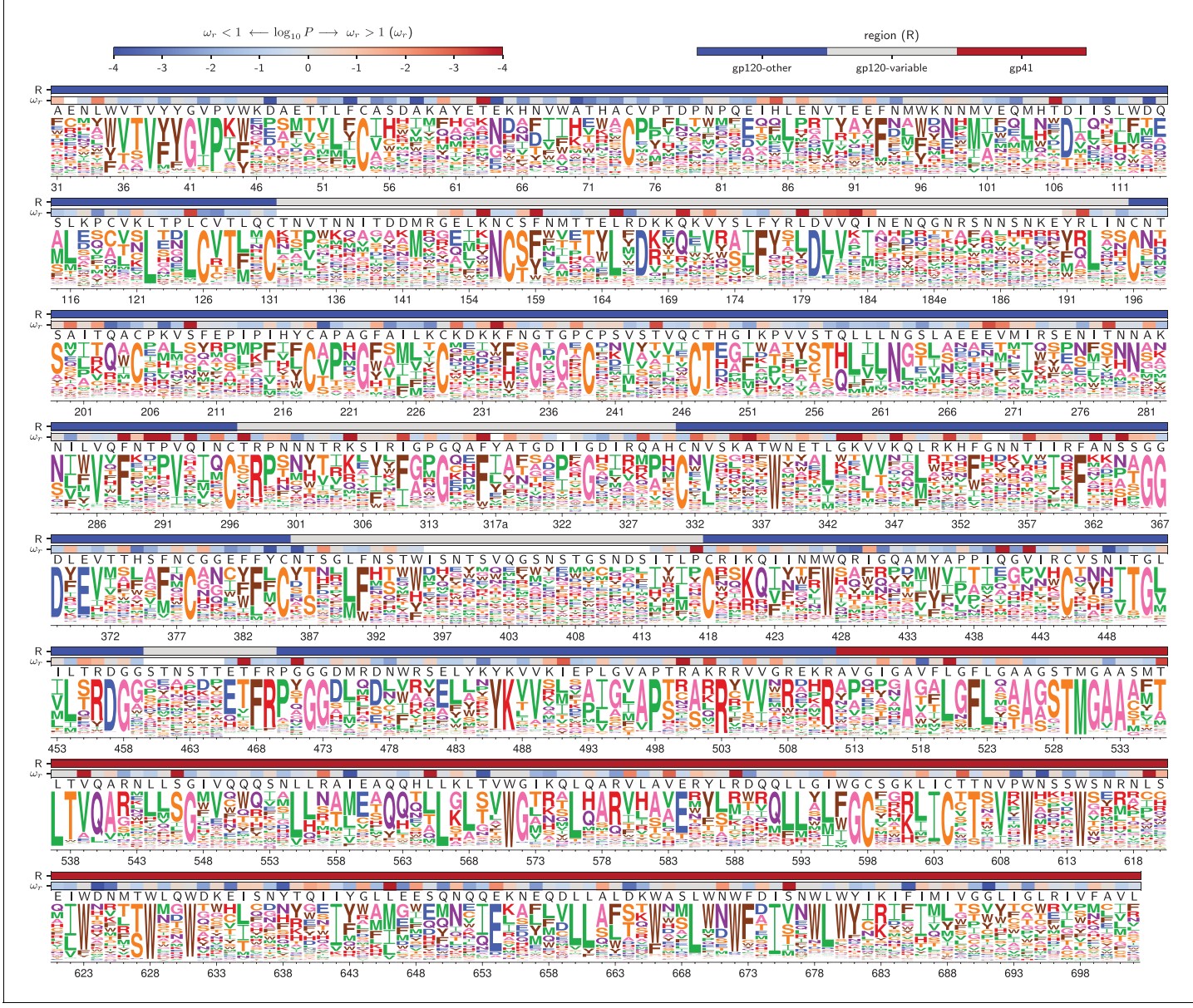

**Figure 4.** Amino acid preferences for the BG505 Env. At each site, the height of the letter is proportional for that site's preference for that amino acid. The top color bar indicates the region of Env (gp120 variable loop, gp120 not variable loop, or gp41). The lower color bar indicates the evidence that the site evolves faster ($\omega_r > 1$) or slower ($\omega_r < 1$) than expected given the experiments (*Bloom, 2017*). We report the p-value for $\omega_r \neq 1$ rather than the value of $\omega_r$ itself since point estimates of $\omega_r$ are unreliable for individual sites due to low numbers of observations, making the p-value a better indicator of the strength of the statistical evidence for faster or slower than expected evolution (*Kosakovsky Pond and Frost, 2005*; *Murrell et al., 2012*). The letters above the logos indicate the wildtype amino acid in BG505. Sites are numbered using the HXB2 scheme (*Korber et al., 1998*). This logo plot shows the site-specific amino acid preferences for BG505 after averaging the replicates and re-scaling by the stringency parameter in *Table 1*. The figure was generated using dms_tools2 (*Bloom, 2015*), which in turn utilizes weblogo (*Crooks et al., 2004*). The numerical values of the preferences are in *Figure 4—source data 1*, the mapping from sequential to HXB2 numbering is in *Figure 4—source data 2*, and the $\omega_r$ values are in *Figure 4—source data 3*.

DOI: https://doi.org/10.7554/eLife.34420.015

The following source data is available for figure 4:

**Source data 1.** The numerical values of the amino acid preferences plotted in this figure are in rescaled_BG505_prefs.csv.
DOI: https://doi.org/10.7554/eLife.34420.016

**Source data 2.** The sequence of BG505 Env and mapping from sequential (*original* column) to HXB2 numbering (*new* column) is in BG505_to_HXB2.csv.
DOI: https://doi.org/10.7554/eLife.34420.017

**Source data 3.** The $\omega_r$ values and associated p-values for BG505 in HXB2 numbering are in BG505_omegabysite.tsv.
DOI: https://doi.org/10.7554/eLife.34420.018

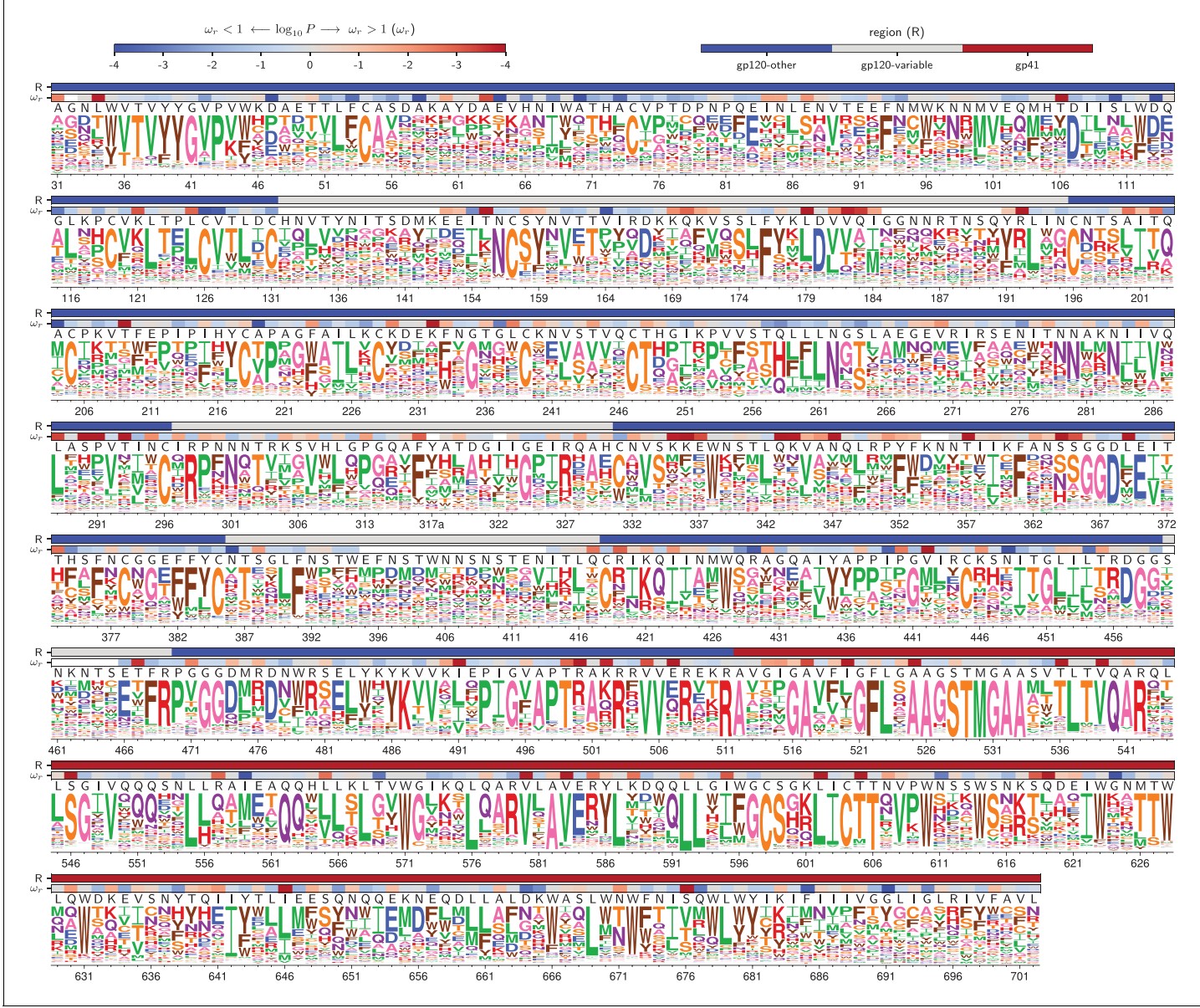

**Figure 5.** Amino acid preferences for the BF520 Env. This figure is the same as *Figure 4* except that it shows the data for BF520 instead of BG505. The numerical values of the preferences are in *Figure 5—source data 1*, the mapping from sequential to HXB2 numbering is in *Figure 5—source data 2*, and the $\omega_r$ values are in *Figure 5—source data 3*.

DOI: https://doi.org/10.7554/eLife.34420.019

The following source data is available for figure 5:

**Source data 1.** The numerical values of the amino acid preferences plotted in this figure are in rescaled_BF520_prefs.csv.

DOI: https://doi.org/10.7554/eLife.34420.020

**Source data 2.** The sequence of BF520 Env and mapping from sequential (*original* column) to HXB2 numbering (*new* column) is in BF520_to_HXB2.csv.

DOI: https://doi.org/10.7554/eLife.34420.021

**Source data 3.** The $\omega_r$ values and associated p-values for BF520 in HXB2 numbering are in BF520_omegabysite.tsv.

DOI: https://doi.org/10.7554/eLife.34420.022

tail reaches its upper value with site 512, which has a shift of 0.52 out of a maximum possible of 1.0. How should we interpret this distribution—have mutational effects shifted a lot, or not very much? We can establish an upper-bound for how much sites might shift by comparing Env to a *non*-homologous protein. *Figure 6B* shows the distribution of shifts when comparing Env to influenza's

hemagglutinin protein, which has previously had its amino acid preferences measured by deep mutational scanning (*Doud and Bloom, 2016*). Most sites have large shifts between Env and hemagglutinin, with the typical shift being ∼0.4 and some approaching the maximum value of 1.0. We can also establish a lower-bound by creating a null distribution for the expected shifts if all differences are simply due to experimental noise. This null distribution is created by randomizing the experimental replicates among Envs. *Figure 6B* shows that the null distribution is more peaked at zero than the real distribution, and does not have the same prominent tail of sites with large shifts. The answer to the question of how much mutational effects have shifted is therefore nuanced: they have substantially shifted at some sites, but remain vastly more similar between the two Envs than between two unrelated proteins.

We can use the null distribution to identify sites where the shifts between BG505 and BF520 are significantly larger than the noise in our experiments (*Figure 6B*). There are 30 such sites at a false discovery rate of 0.1. *Figure 6C* shows the amino acid preferences of these significantly shifted sites for each Env. For the majority of shifted sites, one Env prefers a specific amino acid whereas the other Env tolerates many amino acids; for instance, see sites 512, 516, 599, 165, 605 and 505 in *Figure 6C*. Such broadening and narrowing of a site's mutational tolerance is frequently linked to changes in protein stability, with a more stable protein typically being more mutationally tolerant (*Wang et al., 2002*; *Bloom et al., 2006*; *Gong et al., 2013*; *Kumar et al., 2017*). Work with engineered Env protein in the form of 'SOSIP' trimer (*Binley et al., 2000*; *Sanders et al., 2002*) has shown that BG505 SOSIP is more thermostable than BF520 SOSIP (*Verkerke et al., 2016*). Consistent with this fact, sites with altered mutational tolerance are often (although not always, see sites 165 and 520 in *Figure 6C*) more mutationally tolerant in BG505. Differences in Env's expression level might also contribute to a general broadening or narrowing of tolerance to subsequent mutations. The reason is that our experiments select for viral growth (which is affected by both Env function and expression), so it is possible that some of the shifts are due to epistatic mutational effects on expression rather than function.

However, not all of the significantly shifted sites show a simple pattern of broadening or narrowing of mutational tolerance. For instance, site 288 does not alter its mutational tolerance but rather flips its rather narrow amino acid preference from phenylalanine in BG505 to leucine in BF520 (*Figure 6C*). Thus, there is variation in both the extent and types of shifts observed.

## Structural and evolutionary properties of shifted sites

What distinguishes the sites that have undergone significant shifts? First, we analyzed the distribution of shifted sites in context of Env's three-dimensional structure. Env's structure is highly conformationally dynamic and undergoes large changes upon receptor binding and membrane fusion. In an effort to account for these dynamics, we examined multiple conformational states of Env: the closed pre-fusion state (*Stewart-Jones et al., 2016*), the open CD4-bound state (*Ozorowski et al., 2017*), and the post-fusion six-helix bundle (*Weissenhorn et al., 1997*). *Figure 7A* shows the locations of the shifted sites on the crystal structure of Env in the closed pre-fusion state. There is no visually obvious tendency for shifted sites to preferentially be on Env's surface or in its core, and statistical analysis of both the closed and open states of Env (*Figure 7B*) finds no association between a site's relative solvent accessibility and whether its amino acid preferences have shifted. We did not attempt to analyze the association between solvent accessibility and shift for the post-fusion six-helix bundle because crystal structures of this conformation only contain ∼80 Env residues (*Weissenhorn et al., 1997*; *Chan et al., 1997*; *Tan et al., 1997*).

However, *Figure 7A* does suggest that the sites of significant shifts tend to cluster in Env's structure. A statistical analysis confirms that there is clustering of shifted sites for the closed and open conformations, with the effect being strongest when we define contacts based on the closest intraresidue distance across these two conformations (*Figure 7C*). Therefore, the factors that drive shifts in Env's mutational tolerance often affect physically interacting clusters of residues in a coordinated fashion. We also investigated clustering of shifted sites in the post-fusion six-helix bundle. Because structures of this conformation only resolve the coordinates of ∼80 residues, we did not perform a statistical analysis. However, a qualitative analysis revealed that three of the four shifted sites that are resolved in the post-fusion conformation cluster at one end of the helical bundle (*Figure 7—figure supplement 1*).

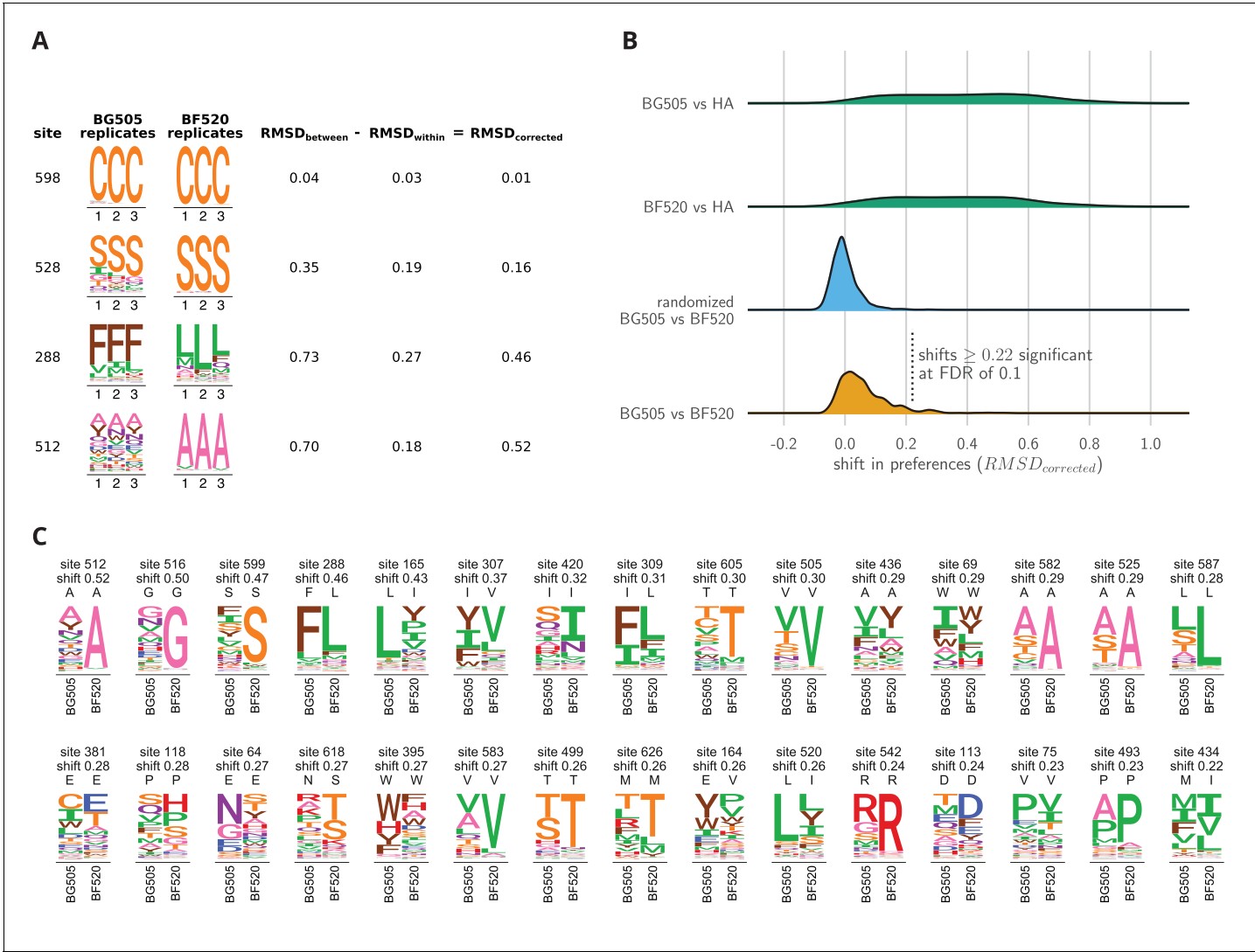

**Figure 6.** Env sites with shifted amino acid preferences between BG505 and BF520. Note that the preferences have been re-scaled using the stringency parameters in *Table 1* to enable direct comparison across Envs. (**A**) Calculation of the corrected distance between the amino acid preferences of BG505 and BF520 at four example sites. We have triplicate measurements for each Env. We calculate the distance between each pair of replicate measurements, and group these into comparisons *between* the two Envs and *within* replicates for the same Env. We compute the root-mean-square distance (RMSD) for both sets of comparisons, which we denote as $RMSD_{between}$ and $RMSD_{within}$. The latter quantity is a measure of experimental noise. The noise-corrected distance between Envs at a site, $RMSD_{corrected}$, is simply the distance between the two Envs minus this noise. (**B**) The bottom distribution (orange) shows the corrected distances between BG505 and BF520 at all alignable sites (see *Figure 6— source data 1* for numerical values). The next distribution (blue) is a null generated by computing the corrected distances on all randomizations of the replicates among Envs. The top two distributions (green) compare Env to the non-homologous influenza hemagglutinin (HA) protein (*Doud and Bloom, 2016*) simply putting sites into correspondence based on sequence number. We compute the p-value that a site has shifted between BG505 and BF520 as the fraction of the null distribution that exceeds that shift, and identify significant shifts at a false discovery rate (FDR) of 0.1 using the method of (*Benjamini and Hochberg, 1995*). Using this approach, 30 of the 659 sites have significant shifts (corrected distance ≥0.22). (**C**) All sites that have significantly shifted their amino acid preferences at an FDR of 0.01. For each site, the logo stacks show the across-replicate average preferences for BG505 and BF520. The wild-type amino acid for that Env is indicated using the small black letters above each logo plot; note how the wild-type amino acid is frequently but not always the most preferred one. The sites are sorted by the magnitude of the shift.

DOI: https://doi.org/10.7554/eLife.34420.023

The following source data is available for figure 6:

**Source data 1.** The corrected distances between BG505 and BF520 at each site are in BG505_to_BF520_prefs_dist.csv.

DOI: https://doi.org/10.7554/eLife.34420.024

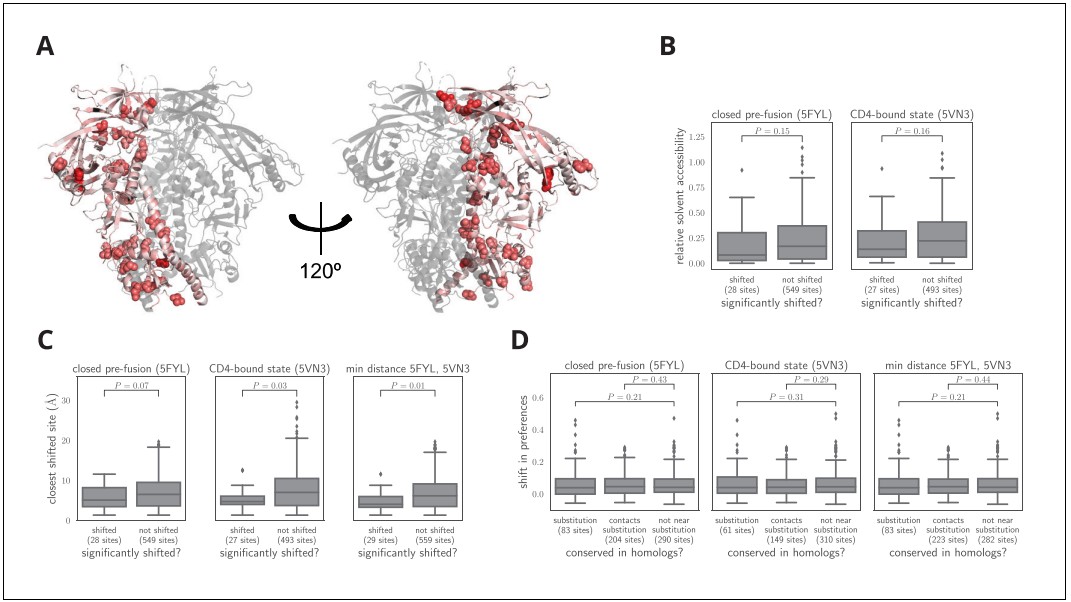

**Figure 7.** Characteristics of significantly shifted sites. (**A**) One monomer of the closed pre-fusion Env trimer (*Stewart-Jones et al., 2016*) is colored from white to red according to the magnitude of the mutational shift at each site (red indicates large shift). Sites that are significantly shifted according to *Figure 6B* are in spheres, and all other sites are in cartoon representation. (**B**) There is no significant difference in the relative solvent accessibility of sites that have and have not undergone significant shifts. This observation holds for both the closed trimer conformation in (**A**) and the CD4-bound trimer conformation (*Ozorowski et al., 2017*). The absolute solvent accessibility of each site was calculated using DSSP (*Kabsch and Sander, 1983*) and normalized to a relative solvent accessibility using the absolute accessibilities from *Tien et al. (2013)*. (**C**) Sites of significant shifts are clustered in the structures of both the closed and open Env trimers. The left two plots show the distance of each significantly shifted and not-shifted site to the closest other shifted site in the indicated structure. The right-most plot shows the minimum distance across both conformations. The trend for shifts to cluster becomes stronger when considering the minimum distance, suggesting multiple conformations contribute to this trend. (**D**) Large mutational shifts are *not* strongly enriched at sites that have substituted between BG505 and BF520, or at sites that contact sites that have substituted. The plots show the magnitudes of the shifts among structurally resolved sites that have substituted between BG505 and BF520, the non-substituted sites that physically contact a substitution in the indicated structure(s) (any non-hydrogen atom within 3.5 angstroms), and all other sites. *Figure 7— source data 1* shows that there is a borderline-significant tendency of significantly shifted sites to have substituted. All plots only show sites that are structurally resolved in the indicated structure(s). Structural distances and solvent accessibilities were calculated using all monomers in the trimer. P-values were calculated using the Mann-Whitney $U$ test. *Figure 7—figure supplement 1* and *Figure 7—figure supplement 2* zoom in on some relevant clusters of sites.

DOI: https://doi.org/10.7554/eLife.34420.025

The following source data and figure supplements are available for figure 7:

**Source data 1.** The sites of significant shifts in *Figure 6B* are somewhat more likely to have substituted between BG505 and BF520.
DOI: https://doi.org/10.7554/eLife.34420.028
**Figure supplement 1.** Cluster of shifted sites in the post-fusion six-helix bundle of Env.
DOI: https://doi.org/10.7554/eLife.34420.026
**Figure supplement 2.** Clusters of shifted sites in highly dynamic regions of Env.
DOI: https://doi.org/10.7554/eLife.34420.027

An obvious hypothesis is that strongly shifted sites have substituted between BG505 and BF520, or physically contact such substitutions. According to this hypothesis, substitutions would alter the local physicochemical environment of the substituted site and its neighbors, thereby shifting the amino acid preferences of sites in the physical cluster. But surprisingly, for both the closed and open conformations, the typical magnitude of shifts is not significantly larger at sites that have substituted, or at sites that contact sites that have experienced substitutions (*Figure 7C*). For the six-helix

bundle, there are five structurally resolved substituted sites, one of which is adjacent to the cluster of shifted sites (*Figure 7—figure supplement 1*). The number of resolved shifted and substituted sites in this structure is too small for a meaningful statistical analysis of the type in *Figure 7D*. However, the cluster of shifted and substituted sites in the six-helix bundle is also present in the closed and open states (*Figure 7—figure supplement 1*), and so is included in the statistical analyses in *Figure 7D*.

There is a borderline trend for the significantly shifted sites to be more likely to have substituted between BG505 and BF520 (*Figure 7—source data 1*), but most shifted sites have not substituted (only 8 of the 30 shifted sites differ in amino acid identity between the two Envs). The lack of strong enrichment in shifts at substituted sites contrasts with previous protein-wide experimental (*Doud et al., 2015*) and simulation-based (*Pollock et al., 2012*; *Shah et al., 2015*) studies of shifting amino acid preferences, which found that shifts were dramatically more pronounced at sites of substitutions. The difference may arise because these earlier studies examined proteins that are fairly conformationally static (absolutely so in the case of the simulations). The fact that Env is extremely complex and conformationally dynamic (*Munro et al., 2014*; *Ozorowski et al., 2017*) may increase the opportunities for long-range epistasis to enable substitutions at one site to shift the amino acid preferences of distant sites.

Indeed, many of the shifted sites cluster within regions of Env that are highly conformationally dynamic. *Figure 7—figure supplement 2* shows the structural context of these clusters in finer detail. One cluster is at the trimer apex where two of Env's variable loops pack against one another and against an adjacent protomer. These interactions are likely involved in regulating the transition between conformational states, and upon CD4 binding, these loops become highly disordered (*Guttman et al., 2014*; *Ozorowski et al., 2017*). Mutations at two of the shifted sites in this cluster (165 and 307) have been shown to cause Env to assume aberrant conformations, suggesting that these sites can strongly modulate Env's dynamics (*Lee et al., 2017*). Strikingly, this cluster of shifted sites may reflect previously observed differences in the conformational dynamics of this regions between these two Envs; the V2 region of BF520 SOSIP trimer is more accessible to deuterium exchange than the BG505 SOSIP trimer (*Verkerke et al., 2016*). The other cluster of shifted sites is near a network of hydrophobic amino acids that has been proposed to help transmit the large-scale conformational change that takes place upon CD4 binding (*Ozorowski et al., 2017*). One of the shifted sites (site 69) overlaps with this network, and mutations at another (site 64) have been shown to strongly modulate the relative stability of the open and closed conformations (*de Taeye et al., 2015*). In total, these two clusters consist of nearly half of the shifted sites (13 out of 30). One hypothesis why so many shifted sites cluster in these regions is that their dynamic nature allows long-range epistatic interactions to be readily propagated between substituted sites and distant shifted sites. It is difficult to discern exactly how these interactions might occur, but there is certainly a trend for sites that are conformationally dynamic to also be sites that show shifts in their amino acid preferences during evolution.

## Entrenchment of substitutions modestly contributes to mutational shifts

One idea that has recently gained support in the protein-evolution field is that substitutions become 'entrenched' by subsequent evolution (*Pollock et al., 2012*; *Shah et al., 2015*; *Starr et al., 2017*). Entrenchment is the tendency of a mutational reversion to become increasingly unfavorable as a sequence evolves. Given two homologs, if there is no entrenchment then the effect of mutating a site in the first homolog to its identity in the second will simply be the opposite of mutating the site in the second homolog to its identity in the first. But if there is entrenchment, then both mutations will be unfavorable, since the site is entrenched at its preferred identity in each homolog.

*Figure 8* shows the distribution of effects for mutating all sites that differ between BG505 and BF520 to the identity in the other Env. As expected under entrenchment, the average effect of these mutations is deleterious—although there are a substantial number of sites where the mutational flips are not deleterious. We can get some sense of the magnitude of the entrenchment by comparing the effects of the BG505↔BF520 mutations to the distribution of effects of all possible amino acid mutations (*Figure 8*). This comparison shows that even unfavorable inter-Env mutational flips are generally more favorable than random amino acid mutations. Therefore, entrenchment occurs for some but not all substitutions that distinguish BG505 and BF520, and the magnitude of

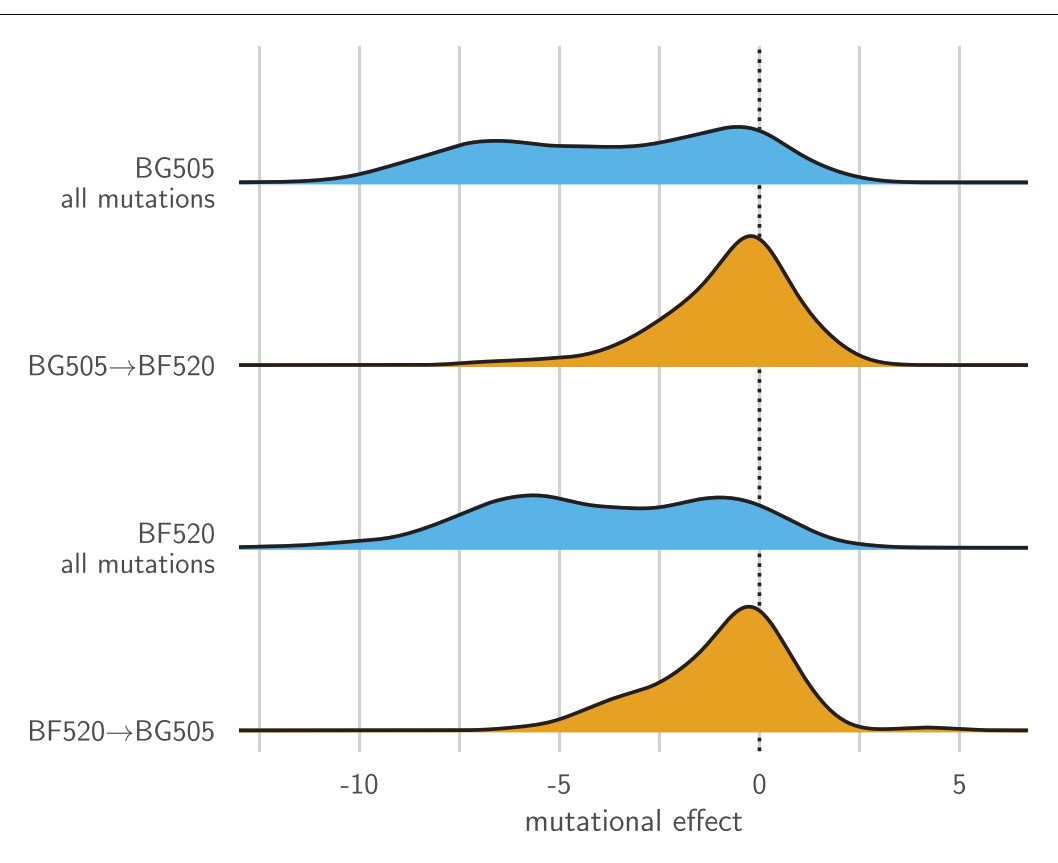

**Figure 8.** Entrenchment of substitutions during Env evolution. There are 12,521 possible amino acid mutations at the 659 mutagenized sites alignable between BG505 and BF520. The blue densities show the effects of all these mutations to each Env. The orange densities show the effects of just the 92 mutations that convert BG505 to BF520 or vice versa. In the absence of entrenchment, mutating a site in BG505 to its identity in BF520 should have the opposite effect of mutating the site in BF520 to its identity in BG505. In this case, we would expect the BF520→BG505 distribution to be the mirror image of the BG505→BF520 distribution—and both distributions should be centered around zero if the two Envs are equivalently functional. Instead, mutating a site in either Env to its identity in the other Env tends to be deleterious, indicating that substitutions are often entrenched in the Env in which they have fixed. The effect of a mutation is quantified as the log of the ratio of the site's preference for the mutant amino acid to the preference for the wild-type amino acid.
DOI: https://doi.org/10.7554/eLife.34420.029

entrenchment is less than the effect of a typical random mutation. Entrenchment of substitutions therefore contributes to some of the mutational shifts. But given that many of these shifts occur at sites that do not even differ between the Envs (*Figure 7D*), entrenchment of substitutions is clearly not the only cause of the shifting amino acid preferences.

## Comparing selection in the lab to natural selection

Our experiments measure the effects of mutations on viral growth in a T-cell line in the lab. But HIV actually evolves in humans, where additional selection pressures on Env are undoubtedly present. For instance, antibody pressure might increase the rate of evolution at some sites (*Albert et al., 1990*; *Wei et al., 2003*; *Richman et al., 2003*), whereas pressure to mask certain epitopes (*Kwong et al., 2002*) might add constraint at other sites. Comparing selection in our experiments to natural selection can identify sites that are under such additional pressures during HIV's actual evolution in humans.

We determined whether each site in Env evolves faster or slower in nature than expected given three models: that evolution is purely neutral (all nonsynonymous and synonymous mutations have equivalent effects), that sites are under the protein-level constraint measured in our experiments

with BG505, or that sites are under the constraint measured with BF520. The first model used a standard dN/dS test (*Kosakovsky Pond and Frost, 2005*), whereas the other two models are conceptually similar but account for the experimentally measured amino acid preferences as described by (*Bloom, 2017*). All three models test if individual sites evolve faster or slower than expected, but they 'expect' different things: the dN/dS model expects nonsynonymous and synonymous mutations fix at the same rate, while the ExpCM expects the rate at a site to depend on the experimentally measured functional constraints. In all cases, the evidence that a site $r$ evolves differently than expected is statistically summarized by the p-value that $\omega_r$ is > or < 1. The standard dN/dS model finds hundreds of sites that evolve slower than expected under neutral evolution (*Table 1*, $\omega_r < 1$), and only a handful of sites that evolve faster than expected under neutral evolution (*Table 1*, $\omega_r > 1$). This finding is unsurprising, since it is well known that Env is under functional constraint. In contrast, ExpCM's that test the rates of evolution relative to the experimentally measured constraints find far fewer sites that evolve slower than expected, but many more sites that evolve faster (*Table 1*).

The sites that evolve slower or faster than expected from the experiments are shown in *Figure 9A, B*, and overlaid on the logoplots in *Figure 4* and *Figure 5* as the $\omega_r$ values. The identified sites are similar regardless of whether we use the experiments with BG505 or BF520 (*Figure 9C*). The reason the results are similar for both experimental datasets is that (as discussed above) the amino acid preferences of *most* sites are similar in both Envs, suggesting that either dataset provides a reasonable approximation of the site-specific functional constraints across the clade A Envs in *Figure 1*.

What causes some sites to evolve faster or slower in nature than expected from the experiments? The answer in both cases is likely to be immune selection. Most of the sites of faster-than-expected evolution are on the surface of Env (*Figure 9A, B* and *Figure 9—figure supplement 1*). Env's escape from autologous neutralizing antibodies often involves amino acid substitutions in surface-exposed regions (*Moore et al., 2009*), including at many of the sites that evolve faster than expected. Since our deep mutational scanning did not impose antibody pressure, sites where substitutions are antibody-driven will evolve faster in nature than expected from the experiments.

Interestingly, immune selection also offers a plausible explanation for the sites that evolve *slower* than expected. In addition to escaping immunity via substitutions at antibody-binding footprints, Env is notorious for employing a range of more general strategies to reduce its susceptibility to antibodies. These strategies include shielding immunogenic regions with glycans (*Wei et al., 2003*; *Stewart-Jones et al., 2016*; *Gristick et al., 2016*) or hiding them by adopting a closed protein conformation (*Kwong et al., 2002*; *Guttman et al., 2015*; *Ozorowski et al., 2017*). Sites that contribute to such general immune-evasion strategies will be under a constraint in nature that is not present in our experiments—and indeed, such sites evolve more slowly than expected from our experiments. For instance, we find very little selection to maintain most glycans in our cell-culture experiments. Of the 21 N-linked glycosylation sites shared between BG505 and BF520, only four are under strong selection to maintain the glycan in our experiments—despite the fact that most are conserved in nature (*Figure 9C* and *Figure 9—figure supplement 2*). This finding concords with prior literature suggesting that these glycans are selected primarily for their role in immune evasion (*Pugach et al., 2004*; *Wang et al., 2013*; *Rathore et al., 2017*). Similarly, a network of sites that help regulate Env's transition between open and closed conformations that have different antibody susceptibilities (*Figure 9D*) also evolve slower in nature than expected from our experiments. Therefore, we can distinguish evolutionary patterns that are shaped by simple selection for Env function from those that are due to the additional complex pressures imposed during human infections.

## Discussion

We have experimentally measured the preference for each amino acid at each site in the ectodomain and transmembrane domain of two Envs under selection for viral growth in cell culture. These amino acid preference maps are generally consistent with prior knowledge about sites that are important for protein properties such as receptor binding or disulfide-mediated stability. However, the main value of these maps comes not from comparing them with prior knowledge, but from the fact that such prior knowledge encompasses just a small fraction of the vast mutational space available to Env. Because Env evolves so rapidly, every study of this protein must be placed in an

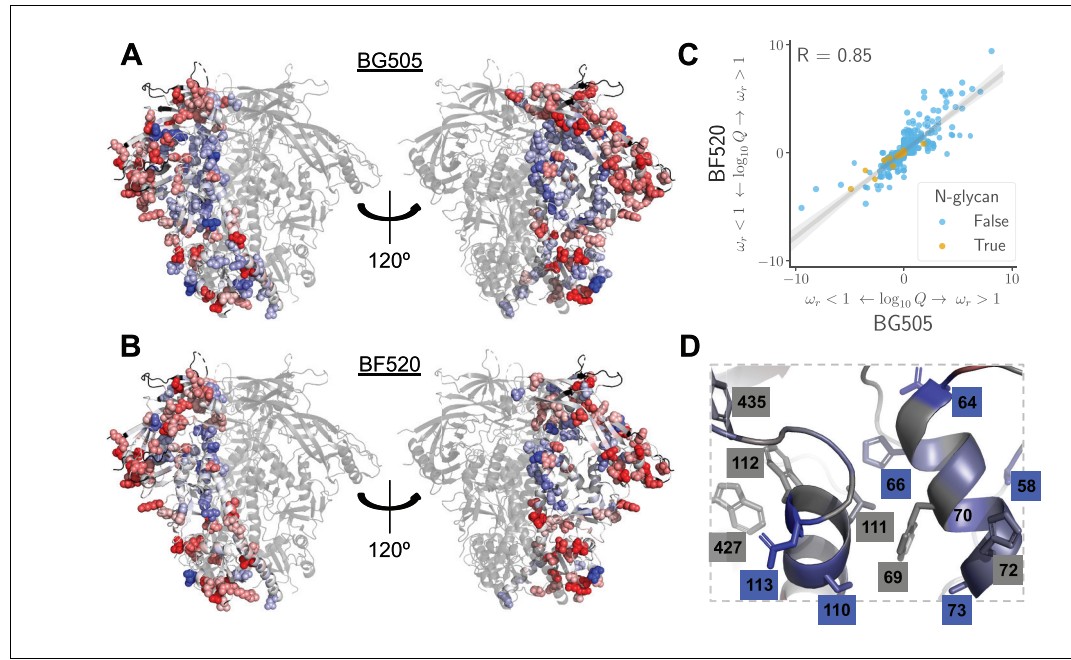

**Figure 9.** Sites in Env that evolve faster or slower in nature than expected given the functional constraints measured in the lab. We calculated the statistical evidence that each site evolves faster ($\omega_r > 1$) or slower ($\omega_r < 1$) than expected given the experimentally measured amino acid preferences using the method of **Bloom, 2017**). (**A**) One monomer of the Env trimer (**Stewart-Jones et al., 2016**) is colored from blue to white to red based on the strength of evidence that sites evolve slower than expected (blue), as expected (white) or faster than expected (red) given the BG505 experiments. Sites for which we lack $\omega_r$ estimates are colored black. Sites where the rate of evolution is significantly different than expected at a false discovery rate of 0.05 are shown in spheres. (**B**) Like (**A**) but using the data from the BF520 experiments. For both Envs, sites that evolve significantly slower or faster than expected are often on Env's surface **Figure 9—figure supplement 1**. (**C**) The results are similar regardless of whether the BG505 or BF520 experiments are used. Many of the sites of slower-than-expected evolution are asparagines in N-linked glycosylation motifs **Figure 9—figure supplement 2**. All sites that evolve slower than expected for both experimental datasets are in **Figure 9—figure supplement 3**. (**D**) A large cluster of sites that evolve slower than expected is likely involved in Env's transition between open and closed conformational states. Gray boxes indicate sites that (**Ozorowski et al., 2017**) proposed form a hydrophobic network that regulates the conformational change; blue boxes and sticks indicate sites that evolve slower than expected. All analyses used the phylogenetic tree in **Figure 1**. The $\omega_r$ and $Q$-values are in **Figure 9— source data 1**.
DOI: https://doi.org/10.7554/eLife.34420.030

The following source data and figure supplements are available for figure 9:

**Source data 1.** The $\omega_r$ and $Q$-values are in merged_omegabysite.csv.
DOI: https://doi.org/10.7554/eLife.34420.034
**Figure supplement 1.** Relative solvent accessibilities of sites evolving faster or slower than expected.
DOI: https://doi.org/10.7554/eLife.34420.031
**Figure supplement 2.** Amino acid preferences and alignment frequencies for glycosylation motifs.
DOI: https://doi.org/10.7554/eLife.34420.032
**Figure supplement 3.** Amino acid preferences and alignment frequencies of sites that evolve slower than expected.
DOI: https://doi.org/10.7554/eLife.34420.033

evolutionary context, and our comprehensive amino acid preference maps potentially enable this in ways that prior piecemeal studies of mutations cannot.

But these maps come with a potentially serious caveat: each one is measured for just a single Env variant. The major question that our study aimed to answer is whether the maps are still useful for evolutionary questions, or whether Env's amino acid preferences shift so rapidly that each map only applies to the specific HIV strain for which it was measured. This question is reminiscent of one that was grappled with in the early days of protein crystallography, when it first became possible to build

maps of a protein's structure. Because it was not (and is still not) possible to crystallize every variant of a protein, it was necessary to determine whether protein structures could be usefully generalized among homologs. Fortunately for the utility of structural biology, it soon became apparent that closely homologous proteins have similar structures (*Chothia and Lesk, 1986*; *Sander and Schneider, 1991*). This rough generalizability of protein structures holds even for a protein as conformationally complex as Env—for although there are many examples of mutations that alter aspects of Env's conformation and dynamics (*Kwong et al., 2000*; *White et al., 2010*; *Almond et al., 2010*; *Davenport et al., 2013*), SOSIP trimer structures from diverse Env strains remain highly similar in most respects (*Julien et al., 2015*; *Pugach et al., 2015*; *Stewart-Jones et al., 2016*; *Verkerke et al., 2016*; *Gristick et al., 2016*).

Our results show that amino acid preference maps of Env also have a useful level of conservation for many purposes. From a qualitative perspective, the amino acid preferences look mostly similar between BG505 and BF520, and so provide a valuable reference for estimating which mutations are likely to be tolerated at each site in diverse HIV strains. Indeed, we anticipate that the complete maps of mutational effects in *Figure 4* and *Figure 5* will be useful for future sequence-structure-function studies. From an analytical perspective, a powerful use of our maps is to identify sites that evolve differently in nature than is required by the simple selection for viral growth imposed in our experiments—and the identified sites are largely the same regardless of whether the analysis uses an amino-acid preference map from BG505 or BF520.

Of course, from the perspective of protein evolution, the most interesting sites are the exceptions to the general conservation of amino-acid preferences. Consistent with studies of other proteins (*Natarajan et al., 2013*; *Harms and Thornton, 2014*; *Doud et al., 2015*; *Starr et al., 2017*), we find a subset of sites that change markedly in which mutations they tolerate. Some shifted sites simply accommodate more amino acids in the more stable BG505 Env—a type of shift that has been well-documented for other proteins (*Wang et al., 2002*; *Bloom et al., 2006*; *Gong et al., 2013*; *Kumar et al., 2017*). But interestingly, there is no strong trend for shifts to be enhanced at sites that differ between BG505 and BF520. Recent studies of protein evolution have focused on the idea that substitutions become 'entrenched' as sites shift to accommodate new amino acids (*Pollock et al., 2012*; *Shah et al., 2015*; *Bazykin, 2015*; *Starr et al., 2017*). Indeed, a prior protein-wide comparison of amino acid preferences across homologs of influenza nucleoprotein found a significant enrichment of shifts at sites of substitutions (*Doud et al., 2015*). But although there is some entrenchment of differences between BG505 and BF520, this is not the major factor behind the shifts in amino acid preferences: the majority of sites that have shifted between BG505 and BF520 actually have the same wild-type amino acid in both Envs even though the preferences have shifted. This rather surprising result might be due to Env's exceptional conformational complexity—mutations can cause long-range alterations in Env's conformation (*Kwong et al., 2000*; *White et al., 2010*; *Almond et al., 2010*; *Davenport et al., 2013*), so it seems plausible that they might also shift mutational tolerance at distant sites. Regardless of the exact mechanism, our large-scale datasets of mutational effects in multiple viral strains should be useful for efforts to computationally parameterize 'fitness landscapes' of Env (*Kouyos et al., 2012*; *Ferguson et al., 2013*; *Mann et al., 2014*; *Barton et al., 2015*; *Louie et al., 2018*).

Our experiments provide highly quantitative data on the mutational tolerance of Env under selection for viral growth in cell culture. These data are amenable to rigorous functional and evolutionary analyses. Here, we have shown how these data can be compared between Envs to identify sites where mutational tolerance shifts with viral genotype, or between experiments and nature to identify sites under different pressure in the lab and in humans. Future experiments that modulate selection pressures in other relevant ways should provide further insight into the forces that drive and constrain HIV's evolution.

## Materials and methods

### Creation of codon-mutant libraries

Our codon mutant libraries mutagenized all sites in *env* to all 64 codons, except that the signal peptide and cytoplasmic tail were not mutagenized. The rationale for excluding these regions is that

they are not part of Env's ectodomain and are prone to mutations that strongly modulate Env's expression level (*Chakrabarti et al., 1989*; *Yuste et al., 2004*; *Li et al., 1994*).

The codon-mutant libraries were generated using the approach originally described in (*Bloom, 2014a*), with the modification of (*Dingens et al., 2017*) to ensure more uniform primer melting temperatures. The computer script used to design the mutagenesis primers (along with some detailed implementation notes) is at https://github.com/jbloomlab/CodonTilingPrimers. For BF520, the three libraries are the same ones described by (*Dingens et al., 2017*). For BG505, we created three libraries for this study. The wild-type BG505 sequence used for these libraries is in *Supplemental file 3*. The BG505 mutagenesis primers are in *Supplemental file 4*.

The end primers for the BG505 mutagenesis were: 5'-tgaaggcaaaactactggtccgtctcgagcagaaga-cagtggcaatgagaga-3' and 5'-gctacaaatgcatataacagcgtctcattctttccctaacctcaggcca-3'. As with BF520, we cloned the BG505 *env* libraries into the *env* locus of the full-length proviral genome of HIV strain Q23 (*Poss and Overbaugh, 1999*) using the high-efficiency cloning vector described in (*Dingens et al., 2017*). For this cloning, we digested the cloning vector with BsmBI, and then used PCR to elongate the amplicons to include 30 nucleotides at each end that were identical in sequence to the ends of the BsmBI-digested vector. The primers for this PCR were: 5'-agataggttaattgagagaa-taagagaaagagcagaagacagtggcaatgagagtgatgg-3' and 5'-ctcctggtgctgctggaggggcacgtctcattctttccc-taacctcaggccatcc-3'. Next, we used NEBuilder HiFi DNA Assembly (NEB, E2621S) to clone the *env* amplicons into the BsmBI-digested plasmids. We purified the assembled products using Agencourt AMPure XP beads (Beckman Coulter, A63880) using a bead-to-sample ratio of 1.5, and then transformed the purified products into Stellar electrocompetent cells (Takara, 636765). The transformations yielded between 1.5 and 3.6 million unique clones for each of the three replicate libraries, as estimated by plating 1:2000 dilutions of the transformations. We scraped the plated colonies and maxiprepped the plasmid DNA; unlike in (*Dingens et al., 2017*), we did not include a 4 hr outgrowth step after the scraping step. For the wild-type controls, we maxiprepped three independent cultures of wildtype BG505 *env* cloned into the same Q23 proviral plasmid. See *Figure 2—figure supplement 1* and *Figure 3A* for information on the average mutation rate in these libraries as estimated by Sanger sequencing and deep sequencing, respectively.

## Generation and passaging of viruses

For BG505, we generated mutant virus libraries from the proviral plasmid libraries by transfecting 293 T cells in three 6-well plates (so 18 wells total per library) with a per-well mixture of 2 $\mu$g plasmid DNA, 6 $\mu$l FuGENE 6 Transfection Reagent (Promega, E269A), and 100 $\mu$l DMEM. The 293 T cells were seeded at $5 \times 10^5$ cells/well in D10 media (DMEM supplemented with 10% FBS, 1% 200 mM L-glutamine, and 1% of a solution of 10,000 units/mL penicillin and 10,000 $\mu$g/mL streptomycin) the day before transfection, such that they were approximately 50% confluent at the time of transfection. In parallel, we generated wildtype viruses by transfecting one six-well plate of 293 T cells with each wildtype plasmid replicate. At 2 days post-transfection, we harvested the transfection supernatant, passed it through a 0.2 $\mu$m filter to remove cells, treated the supernatant with DNAse to digest residual plasmid DNA as in (*Haddox et al., 2016*), and froze aliquots at $-80°$C. We thawed and titered aliquots using the TZM-bl assay in the presence of 10 $\mu$g/mL DEAE-dextran as described in (*Dingens et al., 2017*).

We conducted the low MOI viral passage illustrated in Figure 2A in SupT1.CCR5 cells (obtained from Dr. James Hoxie; *Boyd et al., 2015*). The SupT1.CCR5 cells tested negative for mycoplasma. The SupT1.CCR5 cell line was previously created by engineering the parental SupT1 cell line to express CCR5 (*Boyd et al., 2015*). We used antibody staining followed by flow cytometry to validate that our stock of SupT1.CCR5 cells expressed CCR5, CXCR4, and CD4. There is no validated STR profile for SupT1.CCR5 cells. However, we performed STR profiling on our stock of cells and compared the results to the ATCC SupT1 (ATCC #CRL-1942) reference profile. We found that 11 of 14 alleles plus both amelogenin alleles matched the reference, with no additional mismatched alleles ins the SupT1.CCR5 profile. Given the known instability of lymphoma cell lines (*Inoue et al., 2000*), this level of identity suggests that the SupT1.CCR5 cells are indeed related to the parental SupT1 cells (*Capes-Davis et al., 2013*)

During this passage, cells were maintained in R10 media, which has the same composition as the D10 described above, except RPMI-1640 (GE Healthcare Life Sciences, SH30255.01) is used in the place of DMEM. In addition, the media contained 10 $\mu$g/mL DEAE-dextran to enhance viral infection.

We infected cells with 4 million (for replicate 1) or 5 million (for replicates 2 and 3) TZM-bl infectious units of mutant virus at an MOI of 0.01, with cells at a starting concentration of 1 million cells/mL in vented tissue-culture flasks (Fisher Scientific, 14-826-80). At day one post-infection, we pelleted cells, aspirated the supernatant, and resuspended cell pellets in the same volume of fresh media still including the DEAE-dextran. At 2 days post-infection, we doubled the volume of each culture with fresh media still including DEAE-dextran. At 4 days post-infection, we pelleted cells, passed the viral supernatant through a 0.2 $\mu$m filter, concentrated the virus ~30 fold using ultracentrifugation as described in (*Dingens et al., 2017*), and froze aliquots at $-80°$C. In parallel, for each replicate, we also passaged $2 \times 10^5$ (for replicate 1) or $5 \times 10^5$ (for replicates 2 and 3) TZM-bl infectious units of wildtype virus using the same procedure. To obtain final titers for our concentrated virus, we thawed one of the aliquots stored at $-80°$C and titered using the TZM-bl assay in the presence of 10 $\mu$g/mL DEAE-dextran.

For the final short-duration infection illustrated in *Figure 2A*, for each replicate we infected $10^6$ TZM-bl infectious units into $10^6$ SupT1.CCR5 cells in the presence of 100 $\mu$g/mL DEAE-dextran (note that this is a 10-fold higher concentration of DEAE-dextran than for the other steps, meaning that the effective MOI of infection is higher if DEAE-dextran has the expected effect of enhancing viral infection). Three hours post-infection, we pelleted the cells and resuspended them in fresh media without any DEAE-dextran. At 12 hr post-infection, we pelleted cells, washed them once with PBS, and then used a miniprep kit to harvest reverse-transcribed unintegrated viral DNA (*Haddox et al., 2016*).

The generation, passaging and deep sequencing of BF520 was done in a highly similar fashion, except that we only had a single replicate of the wild-type control. Note that the final passaged BF520 mutant libraries analyzed here actually correspond to the 'no-antibody' controls described in (*Dingens et al., 2017*), but that study did not analyze the initial plasmid mutant libraries relative to these passaged viruses, and so was not able to provide measurements of the amino acid preferences.

## Illumina deep sequencing

We deep sequenced all of the samples shown in *Figure 3A*: the plasmid mutant libraries and wild-type plasmid controls, and the cDNA from the final mutant viruses and wildtype virus controls. In order to increase the sequence accuracy, we used a barcoded-subamplicon sequencing strategy. This general strategy was originally applied in the context of deep mutational scanning by *Wu et al. (2014)*, and the specific protocol used in our work is described in *Doud and Bloom, 2016*) (see also https://jbloomlab.github.io/dms-tools2/bcsubamp.html).

The primers used for BG505 are in *Supplementary file 5*. The primers used for BF520 are in (*Dingens et al., 2017*). The data generated by the Illumina deep sequencing are on the Sequence Read Archive under the accession numbers provided at the beginning of the Jupyter notebook in *Supplementary file 1* and *2*.

## Analysis of deep-sequencing data

We analyzed the deep-sequencing data using the dms_tools2 software package (*Bloom, 2015*, https://jbloomlab.github.io/dms_tools2/, version 2.2.4). The algorithm that goes from the deep-sequencing counts to the amino acid preferences is that described in (*Bloom, 2015*) (see also https://jbloomlab.github.io/dms-tools2/prefs.html). A Jupyter notebook that performs the entire analysis including generation of most of the figures in this paper is in *Supplementary file 1*. An HTML rendering of this notebook is in *Supplementary file 2*. A repository containing all of this code is also available at https://github.com/jbloomlab/EnvMutationalShiftsPaper (*Haddox et al., 2018*, copy archived at https://github.com/elifesciences-publications/EnvMutationalShiftsPaper).

The Jupyter notebooks in *Supplementary file 1* and *2* also contain numerous plots that summarize relevant aspects of the deep sequencing such as read depth, per-codon mutation frequency, mutation types, etc. *Supplementary file 1* also contains text files and CSV files with the numerical values shown in these plots.

Citations are also owed to weblogo (*Crooks et al., 2004*) and ggseqlogo (*Wagih, 2017*), which were used in the generation of the logoplots.

## Alignments and phylogenetic analyses of Env sequences

A basic description of the process used to generate the clade A sequence alignment in tree-source data 1, the alignment mask in tree-source data 1, and the phylogenetic tree in tree are provided in the legend to that figure. An algorithmic description of how the alignment and tree were generated are in *Supplementary file 1* and *2*.

For fitting of the phylogenetic substitution models, we used *Table 1* (*Hilton et al., 2017*, http://jbloomlab.github.io/phydms/, version 2.2.1) to optimize the substitution model parameters and branch lengths on the fixed tree topology intree. The Goldman-Yang (or YNGKP) model used in *Table 1* is the M5 variant described by *Yang et al., 2000*), with the equilibrium codon frequencies determined empirically using the CF3 × 4 method (*Kosakovsky Pond et al., 2010*). For the ExpCM shown in *Table 1*, we extended the models with empirical nucleotide frequencies described in *Hilton et al., 2017*) to also allow $\omega$ to be drawn from discrete gamma-distributed categories exactly as for the M5 model. These ExpCM with gamma-distributed $\omega$ were implemented in *Table 1* using the equations provided by (*Yang, 1994*) (see also http://jbloomlab.github.io/phydms/implementation.html#models-with-a-gamma-distributed-model-parameter). The preferences were re-scaled by the stringency parameters in *Table 1* as described in *Hilton et al., 2017*). For both the M5 model and the ExpCM with a gamma-distributed $\omega$, we used four categories for the discretized gamma distribution.

*Table 1—source data 1* shows the results for a wider set of models than those used in *Table 1*. These include the M0 model of (*Yang et al., 2000*), ExpCM without a gamma-distributed $\omega$, and ExpCM in which the amino acid preferences are averaged across sites as a control to ensure that the improved performance of these models is due to their site-specificity. Note how for these Env alignments, using a gamma-distributed $\omega$ is very important in order for the ExpCMs to outperform the M5 model—we suspect this is because there are many sites of strong diversifying selection.

For detection of sites with faster or slower than expected evolution, we used the approach in (*Bloom, 2017*), which is exactly modeled on the FEL approach of (*Kosakovsky Pond and Frost, 2005*) but extended to ExpCM. This approach estimates a p-value that $\omega_r$ is not equal to one for each site $r$ using a likelihood-ratio test. The actual point estimates of $\omega_r$ are unreliable for individual sites due to the limited number of observations, so we report the p-value that $\omega_r$ is not equal to one, which is a better indication of the strength of the statistical evidence for faster or slower than expected evolution (*Kosakovsky Pond and Frost, 2005*; *Murrell et al., 2012*). For the Q-values and false discovery rate testing, we considered the tests for $\omega_r > 1$ and $\omega_r < 1$ separately.

*Supplementary file 1* and *2* contains the code that runs *Table 1* to reproduce all of these analyses.

## Re-scaling the preferences

The amino acid preferences that are directly extracted from the deep sequencing data essentially give the enrichment/depletion of each mutation, normalized to sum to one at each site (*Doud et al., 2015*, https://jbloomlab.github.io/dms_tools2/prefs.html). However, the extent that any mutation is enriched or depleted is a combination of two factors: the inherent effect of that mutation, and the 'stringency' of the experimental selection. For instance, if the selection is weak, then deleterious mutations will only be slightly depleted; conversely, if selection is strong, then deleterious mutations will be greatly depleted. The fact that the preferences depend on the stringency of the experimental selection is important if we want to compare results between Envs. The reason is that our goal is to identify differences in the inherent effects of mutations between Envs, not simply find differences due to variation in experimental stringency. Of course, we have done our best to perform the experiments for BG505 and BF520 equivalently, but because these are different viruses with different growth rates, it is impossible to exactly match the experimental stringencies. This can be seen in *Figure 3—source data 1*, which shows that stop codons were more depleted for BG505 than BF520, indicating that selection in our experiments was more stringent for BG505.

How should we best re-scale the preferences? Raising them to a power is a sensible approach. To see why, imagine a mutation that is depleted 3-fold after 2 rounds of viral growth. If our experiment instead allowed $2^2 = 4$ rounds of viral growth, then the mutation would be depleted $3^2 = 9$ -fold. More generally, if a mutation is enriched in frequency by $\phi$-fold after $n$ rounds of viral growth, then it will be enriched in frequency by $\phi^{\beta}$-fold after $\beta \times n$ rounds of viral growth. Since the amino acid

preferences are conceptually equivalent to the re-normalized enrichments of mutations (*Bloom, 2015*) https://jbloomlab.github.io/dms-tools2/prefs.html, it therefore makes sense that the re-scaled preference $\pi_{r,a}$ for amino acid $a$ at site $r$ should be related to the directly measured preference $\hat{\pi}_{r,a}$ by $\pi_{r,a} \propto (\hat{\pi}_{r,a})^\beta$. And indeed, this is exactly the re-scaling scheme described in (*Hilton et al., 2017*) that we use to re-scale our preferences for BG505 and BF520.

The last point is how to choose the re-scaling parameter $\beta$ for each Env. It turns out that the features that we have described above for our experiments are also a feature of natural evolution: the expected frequency of a substitution during evolution depends not only on the inherent fitness effect of that mutation, but also on the effective population size, which is conceptually somewhat similar to the stringency of selection. It turns out that in a mutation-selection phylogenetic model of evolution, if the amino acid preferences are taken to represent the 'fitness effects' of mutations, then the exponential scaling parameter $\beta$ is proportional to the effective population size (*Halpern and Bruno, 1998*; *McCandlish and Stoltzfus, 2014*; *Bloom, 2014b*). Therefore, fitting the $\beta$ parameter using a phylogenetic approach enables standardization of the preferences for the two Envs, and re-scales the preferences so that they best match with the actual stringency of selection observed in nature (*Hilton et al., 2017*).

Note that in practice this re-scaling scheme is roughly equivalent to a more heuristic approach that has been used by (*Gray et al., 2017*) and others. In this heuristic approach, the log-transformed enrichment ratios from different experiments are adjusted so that the distributions have equal spreads. Since multiplying log-transformed enrichment ratios is equivalent to exponentiating amino acid preferences, these two re-scaling procedures apply the same mathematical transformation.

## Identifying sites of shifted amino acid preference

When identifying shifts in amino acid preferences between the two Envs, we needed a way to quantify differences between the Envs while accounting for the fact that our measurements are noisy. The approach we use is based closely on that of *Doud et al., 2015*) and is illustrated graphically in *Figure 6A*. The RMSD$_{corrected}$ value is our measure of the magnitude of the shift. *Figure 6A*, its legend, and the associated text completely explains these calculations with the following exception: they do not detail how the 'distance' between any two preference measurements was calculated. The distance between preferences at each site was simply defined as half of the sum of absolute value of the difference between preferences for each amino acid. Specifically, for a given site $r$, let $\pi_{r,a}^i$ be the re-scaled preference for amino acid $a$ in homolog $i$ (e.g. BG505) and let $\pi_{r,a}^j$ be the re-scaled preference for that same amino acid in homolog $j$ (e.g. BF520). Then the distance between the homologs at this site is simply $D_r^{i,j} = \frac{1}{2} \sum_a |\pi_{r,a}^i - \pi_{r,a}^j|$. The factor of $\frac{1}{2}$ is used so that the maximum distance will always fall between zero and one.

## Analysis of entrenchment

For the analysis in *Figuer 8*, the results are presented in terms of the mutational effects rather than the amino acid preferences. If $\pi_{r,a}$ is the preference of site $r$ for amino acid $a$ and $\pi_{r,a'}$ is the preference for amino acid $a'$ (both re-scaled by the stringency parameters in *Table 1*), then the estimated effect of the mutation from $a$ to $a'$ is simply $log\left(\frac{\pi_{r,a'}}{\pi_{r,a}}\right)$.

## Data and code availability

All code and input data required to reproduce all analyses in this paper are in *Supplementary file 1* (see also *Supplementary file 2*). A repository containing all of this code is also available at https://github.com/jbloomlab/EnvMutationalShiftsPaper (*Haddox et al., 2018*; copy archived at https://github.com/elifesciences-publications/EnvMutationalShiftsPaper). The deep sequencing data are on the Sequence Read Archive with the accession numbers listed in *Supplementary file 1* and *2*.

## Acknowledgements

We thank Michael Doud and Orr Ashenberg for computer code that formed the basis for some of the analyses. We thank Andrew Ward for pointing out to us that some of the sites with slower-than-

expected rates of evolution are involved in Env's conformational changes upon receptor binding. We thank Kelly Lee for helpful input about the relative stabilities of the BG505 and BF520 Envs. We thank the Fred Hutch Genomics Core for performing the Illumina deep sequencing.

# Additional information

### Funding

| Funder | Grant reference number | Author |
|---|---|---|
| National Institutes of Health | R01-AI127893 | Jesse D Bloom |
| National Science Foundation | DGE-1256082 | Adam S Dingens |
| Howard Hughes Medical Institute | Faculty Scholar Grant | Jesse D Bloom |
| Simons Foundation | Faculty Scholar Grant | Jesse D Bloom |
| Collaboration for AIDS Vaccine Discovery | OPP1111923 | Jesse D Bloom |
| National Institutes of Health | DP1-DA039543 | Julie Overbaugh |
| National Institutes of Health | T32GM007270 | Hugh K Haddox |

The funders had no role in study design, data collection and interpretation, or the decision to submit the work for publication.

### Author contributions

Hugh K Haddox, Conceptualization, Formal analysis, Investigation, Methodology, Writing—original draft, Writing—review and editing; Adam S Dingens, Conceptualization, Investigation, Methodology, Writing—original draft, Writing—review and editing; Sarah K Hilton, Software, Formal analysis, Writing—review and editing; Julie Overbaugh, Conceptualization, Resources, Supervision, Funding acquisition, Writing—review and editing; Jesse D Bloom, Conceptualization, Software, Supervision, Funding acquisition, Investigation, Methodology, Writing—original draft, Writing—review and editing

### Author ORCIDs

Adam S Dingens (iD) http://orcid.org/0000-0001-9603-9409
Jesse D Bloom (iD) http://orcid.org/0000-0003-1267-3408

### Decision letter and Author response

Decision letter https://doi.org/10.7554/eLife.34420.046
Author response https://doi.org/10.7554/eLife.34420.047

# Additional files

### Supplementary files

• Supplementary file 1. The code to perform all steps in the analysis is in analysis_code.zip. Specifically, this file contains a Jupyter notebook that performs the analysis, all required input data, and all reasonably sized output files. The Jupyter notebook downloads the deep sequencing data, processes it with the dms_tools2 software (*Bloom, 2015*, https://jbloomlab.github.io/dms_tools2/), and also performs a variety of downstream analyses that generate most of the figures for this paper.
DOI: https://doi.org/10.7554/eLife.34420.035

• Supplementary file 2. An HTML rendering of the Jupyter notebook that performs the computational analysis. The actual notebook is in *Supplementary file 1*, but if you just want to look at the analysis rather than run it, then you may prefer this file instead. In particular, the notebook contains plots detailing the deep sequencing data analysis as generated using the dms_tools2 software (*Bloom, 2015*, https://jbloomlab.github.io/dms_tools2/).
DOI: https://doi.org/10.7554/eLife.34420.036

• Supplementary file 3. The sequence of the wildtype BG505 env used in our study is in FASTA format in the file BG505_env.fasta.
DOI: https://doi.org/10.7554/eLife.34420.037

• Supplementary file 4. The sequences of the primers used for the BG505 codon mutagenesis are in the file BG505_codon_mutagenesis_primers.txt.
DOI: https://doi.org/10.7554/eLife.34420.038

• Supplementary file 5. The primers used for the BG505 barcoded-subamplicon sequencing are in the file BG505_bcsubamp_primers.txt.
DOI: https://doi.org/10.7554/eLife.34420.039

• Transparent reporting form
DOI: https://doi.org/10.7554/eLife.34420.040

## Data availability

The following datasets were generated:

| Author(s) | Year | Dataset title | Dataset URL | Database and Identifier |
|---|---|---|---|---|
| Dingens AS, Haddox HK, Overbaugh J, Bloom JD | 2017 | Deep mutational scanning of BF520 | https://www.ncbi.nlm.nih.gov/sra?term=SAMN06313000 | Sequence Read Archive, SAMN06313000 |
| Haddox HK, Dingens AS, Hilton SK, Overbaugh J, Bloom JD | 2017 | Deep mutational scanning of BG505 | https://www.ncbi.nlm.nih.gov/sra?term=SAMN07718028 | Sequence Read Archive, SAMN07718028 |

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
