## [Decision Letter]

Thank you for submitting your article "Mapping mutational effects along the evolutionary landscape of HIV envelope" for consideration by *eLife*. Your article has been reviewed by three peer reviewers, and the evaluation has been overseen by Arup Chakraborty as the Reviewing Editor and Senior Editor. The following individuals involved in review of your submission has agreed to reveal their identity: John P Barton (Reviewer #1); Peter Kim (Reviewer #3).

The reviewers have discussed the reviews with one another and the Reviewing Editor has drafted this decision to help you prepare a revised submission.

Summary:

Information on the possible amino acid trajectories of the rapidly evolving HIV envelope protein has long been limited. In this manuscript, you extend previous work from your lab analyzing the mutational landscape of the HIV-1 Env gene during in vitro viral replication. The work focuses on two clade A transmitter founder viruses, BG505 and BF520, which differ by more than 100 mutations. A deep mutational scanning approach is used to identify the permitted mutations at each residue of Env. Of particular interest is how amino acid preferences differ for these two proteins. A set of 30 sites with clear shifts in amino acid preference are identified. These data are compared to known mutational constraints on Env and to one another, and sources for the shifts in mutant affects are proposed by eliminating many scenarios. The results offer a body of knowledge useful for identifying i) residues and networks that are critical for basic function and ii) hot spots for evolvability or immune pressure. The analysis has been performed to a high standard (though some clarifications are necessary), and the openness of the data is a great service to the community. The paper represents a significant contribution towards understanding the structure-function relationship of HIV envelope. Therefore, the paper is likely to be accepted for publication if the following points are adequately addressed.

Essential revisions:

1) How are mutations that strongly modulate Env's expression level taken into account?

2) Only the native, pre-fusion Env trimer is used to calculate proximity of substituted and shifted residues. It is possible that substitution-shift colocalization may be detected in other conformations of Env as it undergoes structural rearrangement to mediate fusion, as opposed to acting through longer-range allostery in the pre-fusion conformation. Another well-characterized structure that should be considered is the post-fusion six-helix bundle of gp41.

3) Overall, it is found that the shifted sites tend to cluster together, but they are not necessarily the same as the sites with substitutions between the two Envs or ones that contact them. This analysis is not pursued in great detail, however, leaving the reasons why shifted sites tend to cluster together unexplained. Given that the authors also observe some evidence of entrenchment of substitutions between the Envs, is it possible that interactions between the shifted sites themselves are important? This question could be explored through a more detailed examination of the structure.

---

## [Author Response]

Essential revisions:1) How are mutations that strongly modulate Env's expression level taken into account?

This is an important point. Our experiments directly measure how mutations affect viral growth. This approach has many advantages over simple surface display approaches, since it is more relevant to the true function of Env. However, a caveat is that viral growth is a convolution of Env function and expression, so our experiments cannot separate these two phenotypes.

We have taken steps to limit simply finding mutations affect expression. We have done this by *not* mutagenizing the signal peptide and cytoplasmic tail, which are the most common sites of mutations that affect expression levels.

We have added text to make this point clear, and emphasize the associated caveats. When we first describe the deep mutational scanning approach, we now say:

“However, in our experiments we mutagenized only the ectodomain and transmembrane domain of Env, and excluded the signal peptide and cytoplasmic tail. The reason is that we measure how Env mutations affect viral growth, which is influenced both by the functionality of Env protein molecules and their expression level. Mutations in the signal peptide and cytoplasmic tail commonly affect Env expression level (Chakrabarti et al., 1989; Yuste et al., 2004; Li et al., 1994), so we excluded these regions with the goal of reducing the degree to which we simply identified mutations that affected Env expression.”

Then when we describe the results, we clearly emphasize how expression level (like Env protein stability, which was already discussed) could have general effects on the results:

“Differences in Env’s expression level might also contribute to a general broadening or narrowing of tolerance to subsequent mutations. Because our experiments select for viral growth – which is affected by both Env function and expression – it is possible that some of the shifts are due to epistatic mutational effects on expression rather than function.”

2) Only the native, pre-fusion Env trimer is used to calculate proximity of substituted and shifted residues. It is possible that substitution-shift colocalization may be detected in other conformations of Env as it undergoes structural rearrangement to mediate fusion, as opposed to acting through longer-range allostery in the pre-fusion conformation. Another well-characterized structure that should be considered is the post-fusion six-helix bundle of gp41.

This is an excellent suggestion. In the initial paper, we only analyzed the closed prefusion conformation of the Env trimer. We now analyze substitution-shift co-localization in two other conformations: the open CD4-bound conformation and the post-fusion six-helix bundle. In the end, our basic conclusions remain the same, adding greater support to the hypothesis that epistasis is acting via long-range interactions.

First, we investigated the open CD4-bound conformation. Figure 7D statistically tests the hypothesis that shifts are larger at sites that have substituted or sites that contacts substitutions. Initially, this figure only showed the results of this test for the closed prefusion conformation. Now, it shows the results for both the closed and open conformations. Both conformations gave the same general result: there is no significant trend for shifted sites to contact sites of substitutions. We also considered both structures at once, classifying sites as contacting substitutions if they do so in at least one structure. Even then, we did not find an association (see the new right-most panel in Figure 7D). Thus, our investigation of the open conformation re-enforced our initial conclusion that many of the shifts are likely due to long-range, rather than short-range, epistatic interactions.

We also repeated the analyses in Figure 7B and Figure 7C with the open conformation. Our initial conclusions about the solvent accessibility (Figure 7B) and clustering of shifted sites (Figure 7B) also held for the open conformation. Next, we investigated substitution-shift co-localization in the post-fusion six-helix bundle. Only ∼80 residues per Env monomer are resolved in available structures of the six-helix bundle. The structure with largest number of shifted sites (PDB: 1ENV) only contains four shifted sites and five substituted sites. These numbers are too low for the statistical tests used in Figure 7. But we have added a new figure (Figure 7—figure supplement 2) that qualitatively analyzes substitution-shift co-localization in the post-fusion six-helix bundle. Three of the four shifted sites cluster at one end of the helix, along with one of the five substitutions. However, this cluster is also present in the closed and open states as well (Figure 7—figure supplement 2). Thus, the post-fusion six-helix bundle did not reveal any new patterns of co-localization that were not also present in the other states.

We now describe the results of the analyses in the first two paragraphs of the Results subsection “Structural and evolutionary properties of shifted sites”, interleaved with the original text in these paragraphs.

In the process of making these revisions, we discovered a minor bug that affected our calculations of residue-residue distances. We fixed that bug in the revised version. None of the main results change, but some of the quantitative values in the box plots and the corresponding P-values have changed slightly.

3) Overall, it is found that the shifted sites tend to cluster together, but they are not necessarily the same as the sites with substitutions between the two Envs or ones that contact them. This analysis is not pursued in great detail, however, leaving the reasons why shifted sites tend to cluster together unexplained. Given that the authors also observe some evidence of entrenchment of substitutions between the Envs, is it possible that interactions between the shifted sites themselves are important? This question could be explored through a more detailed examination of the structure.

To address this comment, we examined the clusters of shifted sites in greater detail. Two clusters occur within conformationally dynamic regions of Env. One hypothesis as to why shifted sites cluster in these regions is that their conformationally dynamic nature allows long-range epistatic interactions to be propagated between shifted and substituted sites that are distant from one another.

We have added a new figure supplement (Figure 7—figure supplement 3) that provides a fine-grained structural view of each of the clusters. We have also added an entire paragraph detailing these observations at the end of the section in the Results called “Structural and evolutionary properties of shifted sites”. Overall, these additions strengthen the manuscript by providing a better structural intuition for how long-range epistatic interactions might occur in Env.